# MASPO: Joint Prompt Optimization for LLM-based Multi-Agent Systems

**Zhexuan Wang** [1]  **Xuebo Liu** [1 ✉]  **Li Wang**  **Zifei Shan**  **Yutong Wang** [1]  **Zhenxi Song** [1]  **Min Zhang** [1]

## Abstract

Large language model (LLM)-based Multi-agent systems (MAS) have shown promise in tackling complex collaborative tasks, where agents are typically orchestrated via role-specific prompts. While the quality of these prompts is pivotal, jointly optimizing them across interacting agents remains a non-trivial challenge, primarily due to the misalignment between local agent objectives and holistic system goals. To address this, we introduce MASPO, a novel framework designed to automatically and iteratively refine prompts across the entire system. A core innovation of MASPO is its joint evaluation mechanism, which assesses prompts not merely by their local validity, but by their capacity to facilitate downstream success for successor agents. This effectively bridges the gap between local interactions and global outcomes without relying on ground-truth labels. Furthermore, MASPO employs a data-driven evolutionary beam search to efficiently navigate the high-dimensional prompt space. Extensive empirical evaluations across 6 diverse tasks demonstrate that MASPO consistently outperforms state-of-the-art prompt optimization methods, achieving an average accuracy improvement of 2.9. We release our code at https://github.com/wangzx1219/MASPO.

## 1. Introduction

Recent advancements in LLMs (Achiam et al., 2023; Team et al., 2024) have exhibited exceptional capabilities in context understanding, instruction following, and complex reasoning, demonstrating strong performance across various tasks and scenarios. Building upon these foundations, Multi-Agent Systems (MAS) have emerged as a powerful paradigm for solving multi-stage problems. By orchestrating heterogeneous agents (Liang et al., 2024; Wang et al., 2025a; Du et al., 2024; Zhuge et al., 2024) to communicate and collaborate, MAS often surpass the capabilities of single-agent counterparts. Within such systems, the design of agent-specific prompts is critical, as they not only define the distinct roles of each agent but also govern their interaction dynamics and reasoning trajectories. However, despite their critical importance, the joint optimization of these prompts remains a non-trivial challenge. Unlike single-agent scenarios, MAS optimization involves a combinatorial search space where the optimality of one agent's prompt depends intrinsically on the behaviors of others.

Typically, MAS operate through the collaboration of specialized agents. While existing prompt optimization methods typically rely on labeled data to evaluate prompt quality, this paradigm is ill-suited for MAS (Fernando et al., 2024a; Yuksekgonul et al., 2024). In collaborative settings, specific agents may be tasked with intermediate steps, such as reasoning, reflection, or summarization rather than generating the final output. This leads to a severe credit assignment problem. A critical failure mode in MAS is Local-Global Misalignment, where an intermediate agent satisfies its local instructions perfectly but generates outputs that mislead downstream peers, causing system-wide failure. Although recent self-supervised strategies (Xiang et al., 2025) leverage comparative feedback to assess reasoning quality, they remain confined to an isolated scope, failing to capture how local variations propagate to influence global system outcomes. In the context of MAS, recent works (Opsahl-Ong et al., 2024; Zhou et al., 2025) have introduced Bayesian search strategies utilizing Tree-structured Parzen Estimators (TPE). However, these methods are restricted to selecting prompts from a fixed, discrete candidate pool, thereby limiting their capacity for open-ended optimization and fine-grained adjustment. Consequently, there is an urgent need for a robust framework capable of automating prompt generation in dynamic multi-agent environments.

To address these challenges, we propose MASPO, a joint prompt optimization framework tailored for multi-agent environments. MASPO introduces three key innovations. First, to resolve the credit assignment dilemma, we design a multi-granularity joint evaluation mechanism that integrates Local Validity, Lookahead Potential, and Global Alignment,

---

[1]Institute of Computing and Intelligence, Harbin Institute of Technology, Shenzhen, China. Correspondence to: Xuebo Liu <liuxuebo@hit.edu.cn>.

*Proceedings of the 43rd International Conference on Machine Learning*, Seoul, South Korea. PMLR 306, 2026. Copyright 2026 by the author(s).

assessing an agent's utility through its contribution to the entire causal chain rather than isolated outputs. Second, we introduce Misalignment-Aware Sampling, a targeted technique that explicitly mines and injects historical traces where coordination failed despite local success, guiding the optimizer to diagnose and rectify specific interaction breakdowns. Third, regarding the co-adaptation protocol, we implement a coordinate ascent-style strategy augmented with a Beam Refresh mechanism, which ensures stability by realigning the search tree of each agent in real-time to mitigate the non-stationarity caused by peer agents.

Extensive experiments conducted across diverse domains demonstrate that MASPO consistently delivers significant performance gains over existing baselines. Our primary contributions are summarized as follows:

- **Multi-Granularity Joint Evaluation**: We introduce a composite evaluation metric that resolves the credit assignment dilemma in MAS. By synergizing Local Validity, Lookahead Potential, and Global Alignment, our approach captures the full causal impact of an agent within the collaborative chain.

- **Misalignment-Driven Generative Search**: We design a beam search strategy explicitly guided by Misalignment Cases—scenarios where agents fulfill local roles but induce system-wide failure.

- **Adaptive Optimization Dynamics**: We propose a co-ordinate ascent-based scheduling protocol augmented with a Beam Refresh mechanism. These techniques effectively mitigate the non-stationarity inherent in multi-agent interactions, ensuring that each agent adapts to the evolving behaviors of its peers.

## 2. Preliminary

### 2.1. LLM-Based Multi-Agent Systems

Adopting the graph-theoretic perspective from the recent literature (Chan et al., 2024; Jiang et al., 2023; Wu et al., 2023), we formalize the MAS as a directed communication graph $\mathcal{G} = (\mathcal{V}, \mathcal{E})$. Here, $\mathcal{V} = \{v_i\}_{i=1}^{N}$ represents the set of $N$ agents, and the edge set $\mathcal{E} \subseteq \mathcal{V} \times \mathcal{V}$ defines the communication topology. A directed edge $(v_j, v_i) \in \mathcal{E}$ signifies that the output of agent $v_j$ serves as input context for agent $v_i$. We equip each agent $v_i$ with an LLM-based inference function $f_i \in \mathcal{F}$ and, crucially, a specific system prompt $p_i$. We denote the set of all prompts as $\mathcal{P} = \{p_i\}_{i=1}^{N}$, which constitutes the primary learnable parameters in our optimization framework. Given a global task query $q$, the generation process for a specific agent $v_i$ is formulated as:

$$o_i = f_i(p_i, q, \mathcal{C}_i), \quad \text{with} \quad \mathcal{C}_i = \bigoplus_{v_j \in \mathcal{N}_{in}(v_i)} o_j, \quad (1)$$

where $\mathcal{N}_{in}(v_i)$ denotes the set of predecessors of $v_i$. Here, $\oplus$ denotes the concatenation operation applied in a fixed topological order, and $\mathcal{C}_i$ represents the aggregated context. $o_i$ is the resulting output conditioned on the role-specific agent's prompt $p_i$.

### 2.2. Prompt Optimization

Prompt optimization aims to automate the discovery of optimal instructions that maximize the performance of LLMs on downstream tasks. In the context of our defined MAS, this objective extends from optimizing a single string to jointly optimizing the set of role-specific prompts $\mathcal{P}$. Let $\mathcal{D} = \{(q_k, y_k^*)\}_{k=1}^{|\mathcal{D}|}$ be a dataset consisting of input queries and their corresponding ground-truth labels (or reference answers). The execution of the multi-agent system $\mathcal{G}$ on a query $q$, governed by the prompt configuration $\mathcal{P}$, produces a final system response $o_{glob}$. We abstract this complex interaction process as a composite function $\Phi$:

$$o_{glob} = \Phi(\mathcal{G}, \mathcal{P}, q). \quad (2)$$

Here, $o_{glob}$ represents the final output from multiple agents, derived through the topological propagation defined in Eq. (1). The goal of MAS prompt optimization is to identify the optimal configuration $\mathcal{P}^*$ that maximizes the expected performance over the data distribution:

$$\mathcal{P}^* = \underset{\mathcal{P} \in \mathcal{S}^N}{\arg\max} \, \mathbb{E}_{(q, o_{glob}^*) \sim \mathcal{D}} \left[ R\left( \Phi(\mathcal{G}, \mathcal{P}, q), o_{glob}^* \right) \right], \quad (3)$$

where $\mathcal{S}$ denotes the discrete space of natural language strings (prompts), $N$ is the number of agents, and $R(\cdot, \cdot)$ is a scalar scoring function measuring the alignment between the prediction of the system and the ground truth. However, directly optimizing this objective is non-trivial. Since agents fulfill different intermediate roles, the final ground truth $o_{glob}^*$ provides only sparse supervision and does not effectively assign credit to individual steps. To address this, we employ a self-supervised evaluation mechanism as a proxy, which is detailed in Section 3.

This optimization problem presents unique challenges compared to single-agent settings. First, the search space $\mathcal{S}^N$ is combinatorial and high-dimensional. Second, the objective function is non-differentiable with respect to $\mathcal{P}$ due to the discrete nature of language tokens, precluding standard gradient-based updates. Crucially, the agents are functionally coupled: modifying the prompt $p_j$ of an upstream agent $v_j$ alters the input context $\mathcal{C}_i$ for downstream agent $v_i$. This induces a covariate shift in the input distribution that $v_i$ faces, creating a non-stationary optimization landscape that necessitates a joint optimization strategy rather than the independent tuning of individual agents.

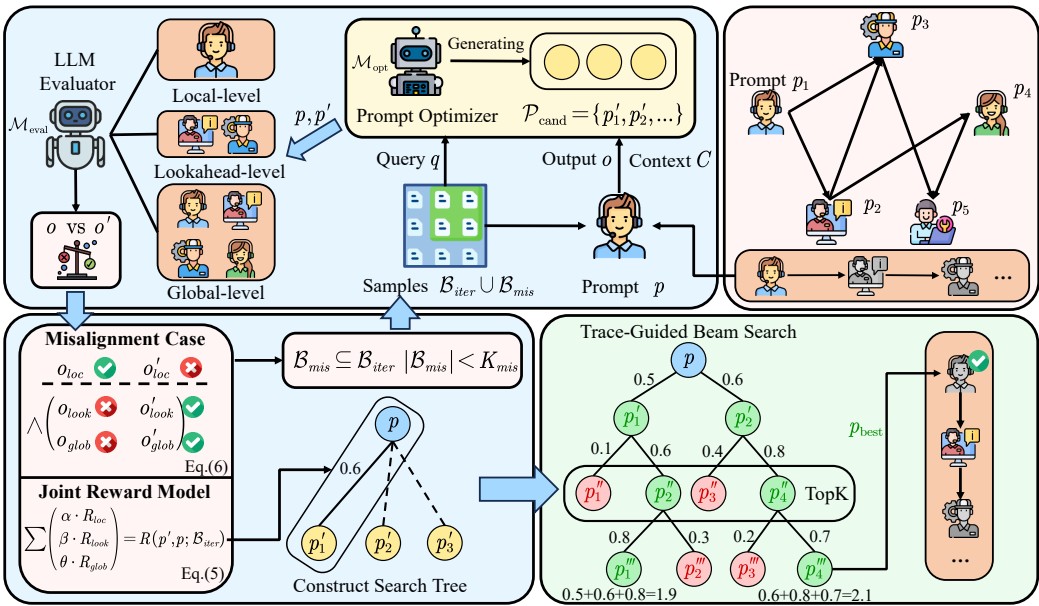

*Figure 1.* Overview of the MASPO Framework. The optimization proceeds sequentially following the topological order of the agent graph (Top-Right). **(Top)** For a specific target agent, the Prompt Optimizer analyzes execution traces (context $\mathcal{C}$ and output $o$) from sampled batches $\mathcal{B}_{iter} \cup \mathcal{B}_{mis}$ to generate candidate prompts $\mathcal{P}_{cand}$. These candidates are rigorously assessed by the LLM Evaluator across three distinct dimensions: local adherence, lookahead potential, and global alignment. **(Bottom-Left)** To resolve credit assignment, we synthesize these evaluations into a Joint Reward Model. Crucially, we identify and mine Misalignment Cases to explicitly guide the optimizer towards repairing coordination breakdowns. **(Bottom-Right)** Navigating the high-dimensional search space, the framework employs a Trace-Guided Beam Search. This mechanism maintains a beam of Top-K candidates, accumulating joint reward scores along the path to iteratively evolve and select the optimal prompt.

## 3. Multi-Agent System Prompt Optimization

We present MASPO, a framework designed to navigate the non-stationary and combinatorial landscape of multi-agent prompt optimization. As illustrated in Figure 1, our workflow follows a systematic loop: it orchestrates agents via a topological protocol, generates candidates through trace analysis, evaluates them using a multi-granularity reward, and evolves the population via an adaptive beam search.

### 3.1. Topological Context and Trace-Guided Proposal

**Topological Scheduling Strategy**  Optimizing the entire MAS simultaneously is intractable due to the functional coupling between agents. To manage this, we adopt a coordinate ascent-style strategy that respects the causal dependencies of the MAS. We iterate through the agents $\{v_1, \dots, v_N\}$ following the topological order of the communication graph $\mathcal{G}$. Unlike standard sequential optimization that fully converges one agent before moving to the next, we employ an interleaved evolution protocol. In each topological turn, we optimize the target agent for a limited number of generations, denoted as the step size $T$, before freezing it and moving to the successor. This process is repeated for $D$ rounds. This interleaved scheduling prevents upstream agents from overfitting to the initial, suboptimal behaviors of downstream peers, thereby stabilizing the co-adaptation process.

**Trace-Guided Generation**  With the target agent fixed, we employ a data-driven generation strategy to explore the prompt space. Unlike blind mutation, our approach grounds offspring generation in actual execution traces. For a parent prompt $p$, we sample a batch $\mathcal{B}_{iter}$ to collect traces $\mathcal{T}_{\text{parent}} = \{(q_k, \mathcal{C}_k, o_k)\}_{k \in \mathcal{B}_{iter}}$. Here, $\mathcal{C}_k$ captures the specific *incoming context*, explicitly modeling the dependency on inter-agent communication. We partition these traces into mini-batches and treat them as few-shot contexts for the Optimizer Model $\mathcal{M}_{\text{opt}}$. $\mathcal{M}_{\text{opt}}$ is instructed to analyze the mapping from input context $(q, \mathcal{C})$ to output $o$, and propose a variation $p'$ that enhances the reasoning logic:

$$\mathcal{P}_{cand} = \bigcup_{m=1}^{K_{sub}} \{p' \mid p' \sim \mathcal{M}_{opt}(p_{\text{parent}}, \tau_m)\}, \quad (4)$$

where $\tau_m$ is a subset of traces. Crucially, to address coordination failures, we employ Misalignment Sampling. We maintain a memory buffer $\mathcal{B}_{mis}$ of "Misalignment Cases" (defined in Sec. 3.2), the scenarios where local validity coexists with ineffective downstream adaptation. During generation, we prioritize injecting $K_{mis}$ samples from $\mathcal{B}_{mis}$. By exposing $\mathcal{M}_{\text{opt}}$ to these hard negatives, we force the generation of offspring that specifically bridge the gap between local instructions and global system goals. The detailed prompt of $\mathcal{M}_{\text{opt}}$ is provided in Appendix A.

## 3.2. Joint Reward Modeling and Misalignment Mining

Once candidate prompts are generated, evaluating their quality presents a severe credit assignment challenge. The output $o_i$ of an upstream agent acts as the input context for downstream agents; thus, relying solely on local validity or final outcome alignment creates an evaluation gap.

**Multi-Granularity Joint Reward** To bridge this gap, we employ a composite scoring function $R(p_{cand}, p_{ref}; \mathcal{B})$ that evaluates the candidate prompt against a reference. The resulting score is a weighted combination of three improvement indicators:

$$
R = \frac{1}{|\mathcal{B}|} \sum_{k \in \mathcal{B}} \left[ \alpha \cdot \underbrace{\mathbb{I}(o_i' \succ o_i)}_{\text{Local Validity}} + \theta \cdot \underbrace{\mathbb{I}(o_{glob}' \succ o_{glob})}_{\text{Global Alignment}} \right.
$$

$$
\left. + \beta \cdot \underbrace{\frac{1}{|\mathcal{N}_{out}(v_i)|} \sum_{v_j \in \mathcal{N}_{out}(v_i)} \mathbb{I}(o_j' \succ o_j)}_{\text{Lookahead Potential}} \right]^{(k)}, \quad (5)
$$

where **Local Validity** measures whether the candidate's output $o_i'$ satisfies role-specific constraints better than the baseline. **Lookahead Potential** is a topology-aware metric that quantifies the "ripple effect" by evaluating whether downstream agents $\{v_j\}$ produce better outputs $o_j'$ when fed with the new context generated by $v_i$, thereby ensuring the prompt produces context useful for immediate successors. **Global Alignment** measures the impact on the final system response $o_{glob}$, capturing long-range dependencies across the entire agent chain. $\mathcal{N}_{out}(v_i)$ denotes the set of immediate successor agents, and $\succ$ represents a preference judgment derived from the Evaluator Model $\mathcal{M}_{eval}$, whose detailed prompt can be found in Appendix B.

**Mining Misalignment Cases** This joint evaluation mechanism allows us to explicitly identify Local-Global Misalignment. A sample $k$ is identified as a misalignment case if the agent satisfies its local objective but fails to support the system:

$$
\mathbb{I}(o_i' \succ o_i)^{(k)} = 1 \quad \text{AND}
$$
$$
\left( \mathbb{I}(\text{Lookahead})^{(k)} = 0 \ \vee \ \mathbb{I}(o_{glob}' \succ o_{glob})^{(k)} = 0 \right). \quad (6)
$$

where $\mathbb{I}(\text{Lookahead})^{(k)}$ equals 1 if the Lookahead Potential (defined in Eq. 5) is 1. These identified cases are stored in $\mathcal{B}_{mis}$ and fed back into the proposal stage (Sec. 3.1) to guide the optimizer in repairing specific interaction breakdowns.

## 3.3. Evolutionary Beam Search with Adaptive Dynamics

To navigate the high-dimensional prompt space efficiently, we integrate the components above into an evolutionary beam search augmented with a dynamic refresh mechanism.

**Trace-Guided Beam Search.** For each optimization step, we maintain a beam of top-$K$ candidates. Each candidate $p' \in \mathcal{P}_{cand}$ is evaluated on $\mathcal{B}_{iter}$, and we calculate the cumulative performance gain by adding the joint reward scores to the parent score:

$$
J(p') = R(p', p_{\text{parent}}; \mathcal{B}_{iter}) + J(p_{\text{parent}}). \quad (7)
$$

This accumulation mitigates the noise of individual samples and enriches the candidate diversity, thereby expanding the search space and allowing us to retain the most robust prompts for the next iteration.

**Beam Refresh Mechanism** A pivotal challenge in MAS optimization is score staleness. Prompts retained in the beam of Agent $v_i$ were evaluated based on contexts generated by obsolete versions of upstream agents. As peer agents evolve during the topological traversal, the input distribution for $v_i$ shifts (covariate shift), rendering historical scores unreliable. To explicitly address this, we discard the stale cumulative scores when an agent is re-visited in a new epoch. We re-anchor the beam by evaluating the relative advantage of each candidate against the current global best prompt $p_{\text{best}}$ (serving as a baseline). We define the refreshed score as the centered win-rate:

$$
J_{new}(p) = R(p, p_{\text{best}}; \mathcal{B}_{iter}) - 0.5, \quad (8)
$$

where subtraction of $0.5$ centers the metric around zero, ensuring that prompts performing worse than the baseline receive negative rewards. By resetting the history, this recalibration ensures the beam search resumes from a valid, up-to-date performance manifold. A detailed description of the algorithm can be found in Appendix C.

## 3.4. Discussion

Recent literature (Wang et al., 2025b; Schmidgall et al., 2025; Xiang et al., 2025) has established that the relative efficacy of prompts can be determined solely by comparing the quality of LLM inference outputs, largely independent of ground-truth labels. Building on this insight, our approach utilizes a small set of unlabeled samples for iterative prompt evolution and evaluation, thereby significantly enhancing practical applicability. In our experiments, we restrict the sample pool to only a few dozen instances. During each optimization iteration, we randomly sample a mini-batch of size $|\mathcal{B}| = 10$ from the pool for trace collection and joint evaluation. This design effectively minimize both the computational overhead and data annotation requirements.

# 4. Experiments

## 4.1. Experimental Setup

**Models and Benchmarks** We conduct experiments using Qwen3-8B (Yang et al., 2025) as the backbone model

*Table 1.* Performance comparison of MASPO against baselines and other optimization methods. **Prompt Opt.** denotes optimizing prompts for individual agents, while **Joint Opt.** indicates the joint optimization of agents within the MAS.

| Method | Prompt Opt. | Joint Opt. | MATH-500 | AGIEval-MATH | AQuA | GPQA | MBPP | HumanEval-ET | Avg. |
|---|---|---|---|---|---|---|---|---|---|
| Vanilla | ✗ | ✗ | 74.80 | 55.86 | 79.53 | 45.96 | 63.47 | 71.95 | 65.27 |
| CoT | ✗ | ✗ | 75.40 | 54.69 | 81.89 | 46.72 | 64.17 | 72.26 | 65.86 |
| SC (CoT) | ✗ | ✗ | 75.50 | 56.64 | 82.17 | 47.52 | 64.49 | 72.56 | 66.48 |
| Self-Refine | ✗ | ✗ | 76.20 | 56.52 | 82.28 | 47.73 | 64.17 | 70.73 | 66.27 |
| AgentDropout | ✗ | ✗ | 76.80 | 59.77 | 86.23 | 47.98 | 58.09 | 72.44 | 66.89 |
| Sequential MAS | ✗ | ✗ | 75.10 | 59.38 | 83.47 | 47.73 | 57.26 | 68.90 | 65.31 |
| + TPE | ✔ | ✗ | 75.80 | 58.73 | 84.92 | 48.04 | 61.30 | 70.12 | 66.49 |
| + SPO | ✔ | ✗ | 77.20 | 60.13 | 81.10 | 49.52 | 63.47 | 67.94 | 66.56 |
| + MASPO | ✔ | ✔ | 77.80 | 61.98 | 85.56 | **58.08** | 65.11 | 73.78 | 70.39 |
| Hierarchical MAS | ✗ | ✗ | 77.60 | 59.38 | 87.01 | 50.63 | 63.93 | 71.34 | 68.32 |
| + TPE | ✔ | ✗ | 77.60 | 60.68 | 86.45 | 49.49 | 64.32 | 71.73 | 68.47 |
| + SPO | ✔ | ✗ | 77.80 | 63.41 | 86.61 | 51.01 | 61.83 | 73.39 | 69.01 |
| + MASPO | ✔ | ✔ | **78.40** | **64.45** | **87.01** | 54.04 | **65.34** | **76.83** | **71.05** |

of MAS, both configured in *standard inference mode* to exclude intrinsic reasoning enhancements. To comprehensively assess system performance, we employ a diverse suite of benchmarks across three domains: (1) **Mathematical Proficiency**: MATH-500 (Hendrycks et al., 2021), AQuA (Patel et al., 2021), and the Level-5 subset of AGIEval-MATH (Zhong et al., 2024); (2) **Complex Reasoning**: the challenging GPQA-Diamond dataset (Rein et al., 2024); and (3) **Code Generation**: MBPP (Austin et al., 2021) and HumanEval-ET (Dong et al., 2025). Furthermore, we utilize Gemini-2.5-pro (Comanici et al., 2025) as the engine for both the optimizer and evaluator modules.

**Baselines** In single-agent scenarios, we compare with the direct reasoning method, known as Vanilla, Chain-of-Thought (CoT, Wei et al., 2022) approach, CoT with self-consistency (SC (CoT), Wang et al., 2023) and Self-Refine (Madaan et al., 2023). For multi-agent collaboration tasks, we establish baselines using two distinct architectures: a Sequential MAS and a Hierarchical MAS (Zou et al., 2025). To ensure a rigorous comparison, we further apply the Tree-structured Parzen Estimator (TPE) used in MIPRO (Opsahl-Ong et al., 2024) and MASS (Zhou et al., 2025) to optimize these MAS configurations. Furthermore, we incorporate SPO as an optimization baseline; despite being a single-agent prompt optimizer, its unsupervised optimization mechanism allows it to be adapted to MAS.

**Implementation Details** For the inference of backbone model of agents, we set the sampling temperature to 0. Regarding the optimization framework, we configure the Optimizer Model $\mathcal{M}_{opt}$ with a temperature of 0.7 to encourage diverse prompt exploration, while the Evaluator Model $\mathcal{M}_{eval}$ operates at a temperature of 0. All models are deployed in non-thinking inference mode. During the iterative optimization phase, we maintain a sample pool of size $|\mathcal{D}| = 50$. The evolutionary search is governed by a beam width of $K = 2$, and for each parent prompt in the

beam, we generate $K_{sub} = 2$ candidate variations. We balance the components of the joint reward model by setting $\alpha = 0.4, \beta = 0.4, \theta = 0.2$. Furthermore, to prioritize error correction while maintaining batch diversity, we set the maximum capacity for retrieved misalignment cases to $K_{mis} = 3$. Regarding the scheduling dynamics, we set the step size for each topological round to $T = 3$, and also set the number of rounds $D = 3$ to ensure that the agent can adapt to the constantly changing cues from its peers. Detailed specifications regarding the initial agent roles, prompt templates, and the architectural configurations for the MAS baselines are provided in Appendix D.

### 4.2. Main Result

**MASPO outperforms other baselines on multiple benchmarks** We observe that MASPO-optimized MAS significantly surpass standard single-agent inference strategies, such as CoT and Vanilla prompting. More importantly, MASPO outperforms heuristic-based collaborative paradigms, including Self-Consistency, Self-Refine, and topological optimization methods like AgentDropout (Wang et al., 2025e). Unlike these static approaches, which rely on fixed role and prompts, MASPO dynamically tailors the interaction logic via prompt evolution, enabling agents to handle intricate dependencies that heuristic methods often overlook. When compared against state-of-the-art prompt optimization techniques, MASPO demonstrates a substantial advantage. While TPE and single-agent adapters SPO provide marginal gains, they often struggle with the non-stationary nature of multi-agent environments. By leveraging joint reward modeling and misalignment-aware sampling, MASPO achieves an average accuracy improvement of 2.90 over the best-performing optimization baselines. This result highlights the efficacy of our method in resolving the credit assignment problem.

**MASPO demonstrates stability across different topologies** We applied MASPO to both Sequential and Hierarchi-

*Table 2.* Comprehensive analysis of MASPO through extensive ablation studies and sensitivity analyses. We examine the framework across eight dimensions, organized into three groups: core mechanism contributions (I–IV), covering the search strategy, scheduling strategy, joint evaluation, and misalignment-aware sampling (with sensitivity to $K_{mis}$); design-choice sensitivities (V–VI), including the lookahead depth and computational budget; and external robustness validations (VII–VIII), assessing the impact of a weaker optimizer/evaluator backbone (Qwen3-8B) and sub-optimal prompt initialization.

| Method | MATH-500 | AGIEval-MATH | AQuA | GPQA | MBPP | HumanEval-ET | Avg. |
|---|---|---|---|---|---|---|---|
| *Reference: Proposed Framework* | | | | | | | |
| **MASPO (Full)** | 77.80 | 61.98 | 85.56 | **58.08** | 65.11 | 73.78 | **70.39** |
| *I. Effectiveness of Search & Scheduling Strategies* | | | | | | | |
| Serial Search | 77.20 | 58.95 | **87.01** | 50.83 | 65.11 | 69.51 | 68.10 |
| Single Cycle | 75.50 | 59.77 | 85.24 | 50.51 | 65.58 | 72.56 | 68.19 |
| Single Agent + SPO | 75.60 | 61.67 | 81.89 | 47.59 | 61.87 | 72.51 | 66.86 |
| + Our Proposed Beam Search | 76.00 | 62.11 | 86.59 | 51.02 | 64.40 | 73.10 | 68.87 |
| *II. Contribution of Core Components* | | | | | | | |
| w/o Beam Refresh | 76.50 | 59.77 | 85.24 | 52.51 | 64.58 | 72.56 | 68.53 |
| w/o Joint Evaluate | 76.20 | 60.13 | 84.85 | 51.01 | 63.70 | 70.73 | 67.77 |
| w/o Misalignment Sampling | 77.60 | 62.89 | 86.61 | 52.53 | 65.28 | 73.17 | 69.68 |
| *III. Sensitivity to Misalignment Cases (Default $K_{mis} = 3$)* | | | | | | | |
| w/ Success-Case Sampling | 77.20 | 61.63 | 86.52 | 51.51 | 65.11 | 73.78 | 69.29 |
| w/ 2 Misalignment Cases | 77.40 | **64.45** | **87.01** | 53.03 | 64.17 | 72.56 | 69.77 |
| w/ 4 Misalignment Cases | 77.60 | 60.13 | 85.83 | 55.41 | 64.64 | 74.73 | 69.72 |
| w/ 5 Misalignment Cases | **78.80** | 60.55 | 86.61 | 51.01 | **67.03** | **75.00** | 69.83 |
| *IV. Robustness to Prompt Initialization* | | | | | | | |
| w/ Minimal Initialization | 77.20 | 62.23 | 86.61 | 56.06 | 64.64 | 72.95 | 69.95 |
| w/ Wrong-Domain Initialization | 77.00 | 61.62 | 85.86 | 55.56 | 65.11 | 73.17 | 69.72 |
| *V. Impact of Lookahead Depth (Default 1-step)* | | | | | | | |
| w/ 2-step Lookahead | 78.00 | 62.33 | 85.86 | 57.58 | 64.04 | 74.73 | 70.42 |
| w/ 3-step Lookahead | 77.80 | 62.65 | 84.85 | 57.07 | 65.11 | **75.00** | 70.41 |
| *VI. Impact of Computational Budget* | | | | | | | |
| SPO + Same Search Budget | 76.80 | 61.33 | 84.85 | 50.51 | 63.70 | 69.52 | 67.79 |
| SPO + Same Gemini Budget | 77.60 | 60.67 | 85.04 | 51.01 | 63.93 | 68.90 | 67.86 |
| *VII. Impact of Optimizer and Evaluator Backbone (Self-Optimized via Qwen3-8B)* | | | | | | | |
| Self-Optimized | 77.00 | 58.92 | 84.58 | 48.48 | 64.64 | 72.56 | 67.70 |

cal MAS structures to assess the architectural adaptability of our framework. Empirical results indicate that our approach is topology-agnostic, yielding performance gains in both settings. Specifically, compared to the respective baselines, MASPO improves the average task accuracy of Sequential MAS by 5.06 and Hierarchical MAS by 2.73. A case study of the optimized prompt is provided in Appendix E.

### 4.3. Analysis

In this section, we conduct a comprehensive series of ablation studies and sensitivity analyses. All experiments in this subsection utilize the Sequential MAS architecture with the Qwen3-8B backbone. The experimental setup remains consistent with the configurations detailed in Section 4.1.

**Effect of Search Strategy** We leverage trace subset guidance combined with a beam search strategy to expand the search space of prompts. Unlike greedy linear search that commits to a single optimization trajectory, our search simultaneously explores multiple promising directions. To validate the contribution of this optimization mechanism, we compare it against a direct linear iterative approach, denoted as "Serial Search" in Table 2. Furthermore, we extend our evaluation to a single-agent scenario to benchmark our

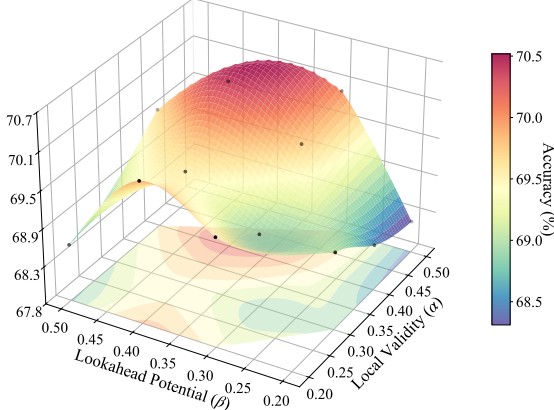

*Figure 2.* Performance landscape of Joint Reward weights. The interpolated surface illustrates the average accuracy with respect to Local Validity ($\alpha$) and Lookahead Potential ($\beta$), under the constraint $\alpha + \beta + \theta = 1$.

method against SPO. In this setting, our framework adapts effectively by retaining the beam search strategy as its core component. The experimental results demonstrate that our search strategy consistently enhances task performance in both single-agent and multi-agent contexts, underscoring the robustness and effectiveness of the proposed method.

*Table 3.* Assessment of cross-model transferability by applying optimized prompts to different model architectures.

| Model | Method | MATH-500 | AGIEval-MATH | AQuA | GPQA | MBPP | HumanEval-ET | Avg. |
|---|---|---|---|---|---|---|---|---|
| Deepseek-V3 | MAS | 78.70 | 66.35 | 85.43 | 55.05 | 68.35 | 76.83 | 71.79 |
| | + Optimized Prompt | 81.50 | 69.14 | 87.40 | 61.11 | 74.94 | 81.10 | 75.86 |
| GLM-4.6 | MAS | 78.80 | 73.05 | 90.16 | 63.13 | 66.79 | 81.71 | 75.61 |
| | + Optimized Prompt | 84.20 | 76.17 | 90.55 | 66.67 | 69.32 | 83.54 | 78.41 |
| Claude-Sonnet-4 | MAS | 84.20 | 74.12 | 89.37 | 64.14 | 71.35 | 82.32 | 77.58 |
| | + Optimized Prompt | 84.50 | 76.95 | 92.17 | 67.14 | 72.83 | 84.76 | 79.73 |
| Gemini-2.5-Pro | MAS | 87.60 | 85.10 | 90.55 | 83.33 | 80.69 | 82.32 | 84.93 |
| | + Optimized Prompt | 87.80 | 86.33 | 92.13 | 85.86 | 83.54 | 87.20 | 87.14 |

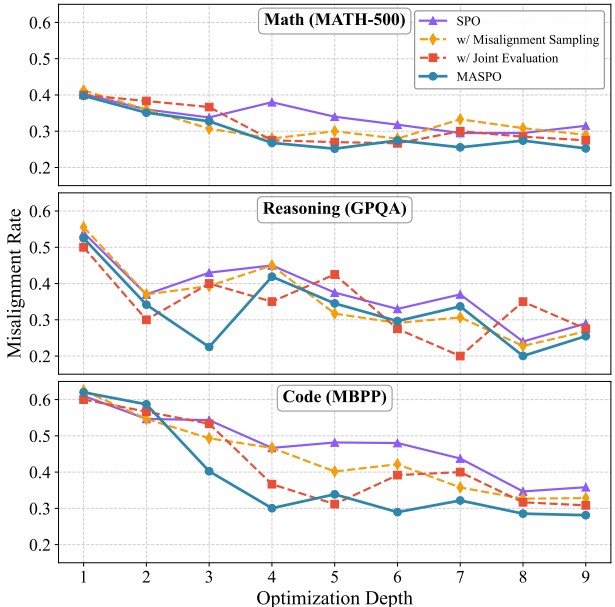

*Figure 3.* Misalignment Rate across optimization depths.

**Effect of Scheduling Strategy**  We employ a coordinate ascent-style scheduling protocol to sequentially optimize each agent. To validate this multi-round iterative co-adaptation strategy, we restrict optimization to a single topological pass ("Single Cycle" in Table 2) under an equal budget, ensuring fair comparison by carefully aligning the total optimization iterations per agent. Furthermore, we assess the criticality of the Beam Refresh mechanism in effectively handling distribution shifts (Row w/o "Beam Refresh"). To mechanistically explain the performance drop in this ablation, we calculated the average Kendall's top-1 overlap of beam candidates' scores between optimization rounds to be only 0.63, revealing that a significant portion of the optimal suggestion candidates change shift and become stale as other agents evolve. The experimental results indicate that both the coordinate ascent scheduling and the beam refresh mechanism are indispensable; their combination consistently enhances overall task performance. This underscores the effectiveness of our strategy in managing the non-stationarity inherent in multi-agent optimization.

**Impact of Joint Evaluation Strategy**  MASPO employs a multi-granularity joint evaluation mechanism to assess non-terminal agents across three distinct dimensions. To validate the efficacy of this composite metric, we assess its contribution via ablation. First, we isolate the evaluation to solely focus on the target agent's intrinsic output quality (i.e., setting lookahead and global weights to zero), denoted as "w/o Joint Evaluate" in Table 2. The performance drop observed in this setting confirms the necessity of holistic evaluation. Furthermore, we conduct a fine-grained sensitivity analysis by varying the weights of the three dimensions. As visualized in Figure 2 (refer to Table 4 in Appendix F for the full numerical data), empirical results indicate that assigning higher weights to local adherence and lookahead potential yields superior adaptability, whereas the optimal weight for global alignment is comparatively lower. Collectively, these results demonstrate that our joint evaluation strategy effectively resolves the credit assignment dilemma, consistently boosting task performance.

**Impact of Misalignment-Aware Sampling**  To bridge the local-global objective gap, MASPO explicitly mines "Misalignment Cases", which are instances that satisfy local roles but fail globally, to guide optimization. Ablation results in Table 2 validate this design: reverting to standard random sampling ("w/o Misalignment Sampling") degrades performance, confirming the inefficiency of unguided exploration. Crucially, a counter-baseline prioritizing successful traces ("w/ Success-Case Sampling") underperforms the random approach. This suggests that merely reinforcing successful behaviors offers low information gain. In contrast, misalignment cases function as hard negatives, providing high-value gradient signals that distinguish genuine global utility from superficial local validity. Sensitivity analysis further indicates that the optimization gain peaks at an injection volume of $K_{mis} = 3$, achieving an optimal trade-off.

To further quantify the effectiveness of our misalignment-aware mechanism, we track the Misalignment Rate, formulated as $\frac{1}{|\mathcal{B}|} \sum_{k \in \mathcal{B}} \mathbb{I}_{\text{mis}}^{(k)}$, where $\mathbb{I}_{\text{mis}}^{(k)}$ indicates whether the sample $k$ satisfies the misalignment criteria defined in Eq. 6. As illustrated in Figure 3, MASPO consistently reduces the misalignment rate as optimization depth increases, and remains lower than other variants. This demonstrates that

explicitly injecting misalignment cases into the optimization loop enables the framework to systematically diagnose and rectify coordination breakdowns, thereby progressively aligning local agent with global system objectives.

**Impact of Lookahead Depth** In our joint reward model (Eq. 5), the Lookahead Potential assesses the immediate impact of a prompt variation on its direct successors (i.e., one-step lookahead). To investigate whether extending this evaluation to multi-hop dependencies would yield further benefits, we conducted experiments evaluating 2-step and 3-step lookahead configurations. As presented in the "w/ 2-step Lookahead" and "w/ 3-step Lookahead" rows of Table 2, deeper lookahead horizons provide negligible performance improvements over the default 1-step setup. This finding strongly confirms that the Global Alignment component integrated into our composite reward function already sufficiently captures complex long-range dependencies and broader system-wide effects without requiring deeper analytical recursion. Consequently, we adopt the one-step lookahead in MASPO to maintain an optimal balance between computational efficiency and overall optimization performance during the execution phase.

**Impact of Computational Budget** To rigorously verify that MASPO's performance gains stem from its inherent algorithmic design rather than merely a higher computational expenditure, we conducted an ablation study controlling for resource allocation. We provided the SPO baseline with expanded resources in two configurations: *SPO + same search budget* scales SPO's total number of optimization steps to match MASPO's, while *SPO + same Gemini budget* ensures SPO consumes an equivalent total number of Gemini API calls as our framework. These two configurations respectively equalize the search depth and the overall inference cost, jointly eliminating resource disparity as a confounding factor. As reported in Table 2, even when granted equivalent computational resources, SPO still substantially underperforms MASPO across the benchmarked evaluation tasks. This ultimately confirms that the superior performance of our framework is fundamentally attributable to its multi-agent-specific optimization mechanisms, specifically the joint evaluation metric and misalignment-aware sampling techniques, rather than relying on brute-force resource scaling or simply increasing query iterations.

**Impact of Optimizer and Evaluator Capabilities** To investigate how the capabilities of the backbone model used for optimization influence the final performance, we conducted an experiment where the agent's own backbone (Qwen3-8B) serves as both the Optimizer ($\mathcal{M}_{opt}$) and Evaluator ($\mathcal{M}_{eval}$), replacing the stronger Gemini-2.5-Pro used in our main setup. As presented in Table 2, MASPO still delivers consistent gains over the vanilla MAS baseline even when driven by this weaker model, indicating that our framework does not critically depend on access to a top-tier optimizer. While employing a more capable foundation model facilitates superior optimization results, the ability of the 8B model to self-improve demonstrates the robustness of our framework, making MASPO applicable to resource-constrained or privacy-sensitive deployments. In addition, further analyses are provided in Appendix G and H.

**Robustness to Prompt Initialization** To verify the effectiveness of MASPO under sub-optimal initialization conditions, we evaluate the framework against two challenging scenarios: (1) Minimal Initialization, where all agent prompts are replaced with a minimal, non-informative instruction ("Answer the question:") lacking any task-specific guidance; and (2) Wrong-Domain Initialization, where initial prompts are deliberately misaligned with the target tasks (e.g., assigning math prompts to code tasks, or code prompts to reasoning tasks). These two settings respectively simulate the absence of prior knowledge and the presence of misleading priors, representing the most adverse starting conditions a practitioner might encounter. As reported in Table 2, both degraded initializations incur only marginal performance drops compared to the default MASPO, while still substantially outperforming all baseline methods. This confirms that MASPO can effectively recover from low-quality or irrelevant initializations, showcasing strong robustness to the quality of user-provided initial prompts.

**Transferability and Robustness Analysis** To strictly evaluate the generalization capability and robustness of our optimization framework, we conducted two sets of extension experiments. We assessed the cross-model transferability of the optimized prompts. Specifically, we directly migrated the role-specific prompts optimized on the Qwen3-8B to distinct agent architectures, including DeepSeek-V3 (Liu et al., 2024), GLM-4.6 (Zeng et al., 2025), Claude-Sonnet-4 and Gemini-2.5-pro, without further fine-tuning. As presented in Table 3, the prompts derived via MASPO yield consistent performance improvements on these unseen backbones. This suggests that MASPO captures generalized interaction logic and reasoning patterns rather than overfitting to the idiosyncrasies of a specific source model. This finding highlights a cost-effective optimization paradigm: when deploying MAS with computationally expensive large-scale models, practitioners can utilize smaller, more efficient models as proxies for the optimization phase.

## 5. Related Work

### 5.1. LLM-based MAS

Research on autonomous systems has transitioned from classical multi-agent reinforcement learning theories (Park et al.,

2023; Yang et al., 2024; Li et al., 2025; Tan et al., 2025) to the modern paradigm of LLM-based MAS. By leveraging the advanced instruction-following and reasoning capabilities of LLMs, these systems enable agents to collaborate on complex planning and problem-solving tasks (Tao et al., 2024; Wang et al., 2025d; Zhao et al., 2025). Foundational frameworks such as ReAct (Yao et al., 2023), AutoGen (Wu et al., 2024), and CAMEL (Li et al., 2023) pioneered this direction by coordinating agents through explicit dialogue, role assignment, and structured communication protocols (Yan et al., 2025; Ye et al., 2025). To enhance collective intelligence, researchers have explored diverse interaction mechanisms, such as multi-agent debate (Liang et al., 2024; Du et al., 2024) and emergent specialization (Mieczkowski et al., 2025; Huang et al., 2025). These collaborative strategies have shown applicability across a wide spectrum of domains, including software development and specialized reasoning in math and science (Pezeshkpour et al., 2024; Yue et al., 2024; Wang et al., 2025c).

As the scale and diversity of agents increase (Wang et al., 2025a; junyou li et al., 2024; Chen et al., 2025a), and as automated construction methods such as AutoAgent (Chen et al., 2024a) and AgentInit (Tian et al., 2025) make it easier to instantiate large-scale MAS, the challenges of computational cost and communication efficiency have become increasingly pronounced (Chen et al., 2024b; Li et al., 2024). Consequently, recent literature has shifted focus toward system-level optimization. Approaches such as Optima (Chen et al., 2025b), AgentPrune (Zhang et al., 2025c) and AgentDropout (Wang et al., 2025e; 2026) seek to refine the communication graph, while dynamic routing mechanisms like MasRouter (Yue et al., 2025), EvoFlow (Zhang et al., 2025a), and MaAS (Zhang et al., 2025b) adaptively select backbone models and sample architectures to balance performance and efficiency. Despite these structural advancements, the joint optimization of the instructional prompts that govern these agent interactions remains a critical yet underexplored challenge.

### 5.2. Prompt Optimization

Prompt optimization aims to automate the design of instructions to maximize LLM performance, serving as a scalable alternative to labor-intensive manual engineering. Approaches generally fall into two categories: continuous soft-prompt tuning and discrete text optimization. While soft-prompt methods utilize gradient-based updates, they often suffer from poor interpretability and are inapplicable to black-box APIs (Cui et al., 2025). Consequently, recent research has shifted toward discrete optimization, employing strategies such as evolutionary algorithms (Guo et al., 2025; Fernando et al., 2024b; Guo et al., 2024), beam search refinement (Chen et al., 2024c; Wang et al., 2024), and gradient-free feedback mechanisms (Yüksekgönül et al.,

2024; Deng et al., 2026). Despite their success, these methods predominantly focus on single-agent scenarios.

Optimizing prompts for MAS presents a significantly more complex challenge due to the intricate coordination required between agents. As the number of agents scales, the interaction space grows combinatorially. Early efforts in this domain, such as GPTSwarm (Zhuge et al., 2024) and MASS, have attempted to co-evolve agent roles and interaction graphs. A primary bottleneck is the evaluation mechanism, which drives the optimization process. Standard frameworks rely heavily on ground-truth labels for benchmark-based assessment (Zhou et al., 2023) that still depend on reference answers. While some works explore consistency-based (Zhang et al., 2024) or human-preference criteria (Lin et al., 2024), efficient and scalable evaluation remains challenging. In contrast to these paradigms, our approach introduces a joint optimization framework designed for MAS, addressing the distribution shift problem.

## 6. Conclusion

In this paper, we introduced MASPO, a novel framework designed to automate the iterative joint optimization of prompts within MAS. Our approach addresses the intrinsic challenges of inter-agent coordination and credit assignment by integrating three core mechanisms: (1) a generative strategy guided by execution traces and misalignment cases to explicitly rectify interaction breakdowns; (2) a multi-granularity joint evaluation metric that resolves the local-global objective gap; and (3) an evolutionary beam search orchestrated via a coordinate ascent protocol with beam refreshment to ensure stable co-adaptation in non-stationary environments. Extensive empirical results validate that MASPO consistently achieves significant and robust performance improvements across diverse reasoning and coding domains. Beyond immediate gains, we believe this framework offers valuable insights for future research into dynamic role specialization and architectural optimization in collaborative AI systems tackling complex tasks.

## Impact Statement

This paper presents a framework for automating prompt optimization in multi-agent systems, with the primary goal of advancing Machine Learning capabilities in collaborative tasks. While the deployment of autonomous agents carries general societal implications regarding automation and safety, the consequences of this work largely depend on the specific applications for which these systems are deployed and the safety properties of the backbone Large Language Models. We do not foresee specific ethical issues or negative societal consequences unique to this optimization framework that require further highlighting.

## Acknowledgments

This work was supported in part by the Guangdong Basic and Applied Basic Research Foundation (Grant No. 2026B0101100004), and Shenzhen Science and Technology Program (Grant No. KJZD20231023094700001, KQTD2024072910215406).

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

# A. Optimizer Prompts

---

**Meta-Optimizer Prompt (Math)**

```
You are optimizing a prompt for a specific agent in a multi-agent mathematical
reasoning system.
CRITICAL: The agent's core role and responsibilities MUST be preserved in the
optimized prompt.

Agent Type:  {agent_type}
Current System Role:  {role_description}

Sample Execution Traces (Question + Context + Agent Output)
```
{samples}
```
Requirements:
```
{requirements}
```
Reference prompt:
```
{prompt}
```
Provide your analysis, optimization points, and the complete optimized prompt using
the following XML format:
<analyse>Analyse what drawbacks exist in the results produced by the reference prompt
and how to improve them.</analyse>
<modification>One sentence summary of the key improvement</modification>
<prompt>Provide the complete optimized prompt</prompt>
```

---

**Meta-Optimizer Prompt (Reasoning)**

```
You are optimizing a prompt for a specific agent in a multi-agent problem-solving
system.
CRITICAL: The agent's core role and responsibilities MUST be preserved in the
optimized prompt.

Agent Type:  {agent_type}
Current System Role:  {role_description}

Sample Execution Traces...
{samples}
Requirements...
{requirements}
Reference prompt...
{prompt}

Optimization Principles:
**DO:**
- Give clear, actionable guidance in 2-3 sentences
- Focus on WHAT to think about, not HOW to format
- The prompt must be CONCISE (under 250 words) and DOMAIN-AGNOSTIC
**DON'T:**
- Create multi-section templates with headers
- Require "verify each step" or "solve independently first"

Provide your analysis, optimization points, and the complete optimized prompt using
the following XML format:
<analyse>...</analyse>
<modification>...</modification>
<prompt>...</prompt>
```

**Meta-Optimizer Prompt (Code)**

```
You are optimizing a prompt for a specific agent in a multi-agent code generation
system.
CRITICAL: The agent's core role and responsibilities MUST be preserved in the
optimized prompt.
Agent Type: {agent_type}
Current System Role: {role_description}

Sample Execution Traces...
{samples}
Requirements...
{requirements}
Reference prompt...
{prompt}

Your Task:
1.  Analyze the "Agent Output" in the samples above.
2.  Identify specific **coding errors** (e.g., off-by-one errors, wrong library usage,
syntax errors) or **format violations**.
3.  Determine if the current prompt is too vague, causing these errors.
4.  Propose a new prompt that explicitly instructs the agent to avoid these specific
pitfalls.

Format your response exactly as follows:
<analyse>Detailed analysis of the bugs found...</analyse>
<modification>One sentence summary...</modification>
<prompt>The complete, optimized prompt (keep it under 300 words).  Ensure it retains
the core task description.</prompt>
```

## B. Evaluator Prompts

**Local and Lookahead Evaluation Prompt (Math)**

```
You are comparing two intermediate outputs from an agent in a multi-agent mathematical
reasoning system.
Your task is to decide which output is more likely to help the system eventually
produce the CORRECT FINAL ANSWER.

Prefer the output that:
- Contains mathematically accurate reasoning (no factual/logical errors)
- Clearly states intermediate results or assumptions
- Avoids misleading statements or ambiguous conclusions
- Provides enough detail for downstream agents to verify or build upon

Problem: {question}
Output A:
{output_a}
Output B:
{output_b}

Which output is more conducive to obtaining the correct final answer?  Respond ONLY
with "A" or "B".
```

---

**Global Evaluation Prompt (Math)**

```
You are comparing two FINAL answers to a math problem.
Your SOLE task is to determine which answer is more likely to be **mathematically
correct**.

Do NOT favor longer, more detailed, or better-formatted responses unless they are also
correct.
If one answer is clearly correct and the other is wrong, choose the correct one|even
if its reasoning is minimal.
If both seem plausible, prefer the one with clearer, error-free reasoning.

Problem:  {question}
Requirement:  {requirement}

Answer A:
{Answer_A}

Answer B:
{Answer_B}

Respond with only "A" or "B".
```

---

**Local and Lookahead Evaluation Prompt (Reasoning)**

```
You are comparing two intermediate outputs from an agent in a multi-agent
problem-solving reasoning system.
Your task is to decide which output is more likely to help the system eventually
produce the CORRECT FINAL ANSWER.

### Evaluation Priority:
**1.  Completeness:**
- Is the output truncated (ends mid-sentence, incomplete thought)?
- If one is truncated and one is not, select the complete one.
**2.  Reasoning Correctness (If both complete):**
- Are scientific facts and principles correctly stated?
- Is the logical reasoning valid?
**3.  Usefulness for Next Agent:**
- Does it provide clear intermediate conclusions?

Problem:  {question}
Output A: {output_a}
Output B: {output_b}

Which output is more conducive to obtaining the correct final answer?  Respond ONLY
with "A" or "B".
```

**Global Evaluation Prompt (Reasoning)**

```
You are comparing two FINAL answers to a problem.
Your SOLE task is to determine which answer is more likely to be **correct**.

### Evaluation Process:
**Step 1:  Check Completeness**
- Does each output have a clear final answer?
**Step 2:  Extract and Compare Answers (if both complete)**
- Identify the final letter choice from each output
**Step 3:  Evaluate Reasoning Quality (if answers differ)**
- Which output correctly applies relevant scientific principles?
- Which reasoning chain is more logically sound?
**Step 4:  Make Decision**
Priority:  Complete > Truncated
Among complete outputs:  Correct reasoning > Incorrect reasoning > Better format

Problem:  {question}
Requirement:  {requirement}
Answer A: {Answer_A}
Answer B: {Answer_B}
Respond with only "A" or "B".
```

**Local and Lookahead Evaluation Prompt (Code)**

```
You are comparing two outputs from a 'Predictor' agent in a code generation pipeline.
The next agent in the pipeline is a 'Reflector' or 'Debugger' that will try to fix any
errors.

Your goal is to select the output that provides the **best starting point** for the
Reflector.

Criteria for selection (in order of importance):
1.  **Correctness**:  If one output appears to be functionally correct python code
while the other has logic errors, choose the correct one.
2.  **Adherence to Requirements**:  The code must NOT contain exception handling...
3.  **Format**:  The code must be easily extractable...
4.  **Logic Clarity**:  If both have bugs, choose the one with the clearer algorithmic
logic that is easier to debug.

Problem:  {question}
Output A: {output_a}
Output B: {output_b}

Which output is better?  Respond ONLY with "A" or "B".
```

---

**Global Evaluation Prompt (Code)**

```
You are comparing two FINAL Python code solutions.
Your SOLE criterion is **Functional Correctness**.

Problem: {question}
Requirement: {requirement}
Answer A: {Answer_A}
Answer B: {Answer_B}

Evaluation Steps:
1.  Mentally trace both codes with edge case inputs.
2.  If Answer A is correct and Answer B is buggy (infinite loop, wrong logic, syntax
error), choose A.
3.  If Answer B is correct and Answer A is buggy, choose B.
4.  If both are correct, choose the one that is **more concise** and follows standard
Pythonic practices.
5.  If both are buggy, choose the one that is "less broken" (closer to the solution).

Respond with only "A" or "B".
```

---

## C. Optimization Algorithm of MASPO

The overall optimization procedure of MASPO is summarized in Algorithm 1. Our framework adopts a topological coordinate ascent strategy to iteratively refine the prompts of agents in the graph $\mathcal{G}$. The process is structured into two distinct phases within each epoch. In Phase 1 (Beam Refresh), specifically starting from the second epoch, we address the non-stationarity issue caused by updates in upstream agents. Before further optimization, we re-evaluate the retained candidates in the beam $B_i$ against the current global best prompt $\mathcal{P}_i$ using a fresh validation batch (Lines 7-15). The scores are recalibrated to centered win-rates (Eq. 8) to ensure the search resumes from an accurate performance manifold reflecting the current input distribution. In Phase 2 (Misalignment-Guided Evolutionary Search), we execute the prompt evolution loop. A crucial step here is the construction of hybrid batches $\mathcal{B}_{iter}$ (Lines 20-22), which mix random samples with hard negatives from the Misalignment Buffer $\mathcal{B}_{mis}$. This forces the optimizer to explicitly target historical coordination failures. Throughout the search rounds $T$, candidates are generated, evaluated via the joint reward model (Eq. 5), and accumulated into the beam. Finally, following a Gauss-Seidel update scheme, any candidate that outperforms the current baseline is immediately synchronized to the global configuration $\mathcal{P}$ (Lines 36-38), ensuring that subsequent agents in the topological order optimize against the most up-to-date context.

## D. MAS Initialization

In this section, we elaborate on the topological configurations and initialization strategies for the MAS baselines employed in our experiments.

**Sequential MAS**  The Sequential MAS is designed to emulate a linear, iterative refinement process. Its topology is structured as an alternating chain of generation and critique, denoted as Predictor → Reflector → Predictor → Reflector. In this architecture, a *Predictor* agent generates an initial solution, which is subsequently analyzed by a *Reflector* agent to identify logical fallacies or syntax errors. The Reflector's feedback serves as the context for the next Predictor in the chain to produce a refined output. This sequential dependency ensures that downstream agents can leverage the reasoning history of upstream agents to progressively improve solution quality.

**Hierarchical MAS**  For the Hierarchical MAS baseline, we adopt a layered structure that emphasizes parallel generation followed by high-level synthesis. We strictly adhere to the topological settings and architectural design proposed in (Zou et al., 2025). This setup employs agent-based solutions for different domain tasks in the first layer, followed by an aggregation agent in the second layer that aggregates the solutions from the agents handling different tasks. -

**Initial Prompts**  All agents within these architectures are initialized with role-specific system prompts tailored to the task type (e.g., Math, Reasoning, or Code Generation).

---

**Algorithm 1** MASPO: Multi-Agent System Prompt Optimization

---

1: **Input:** Graph $\mathcal{G}$, Dataset $\mathcal{D}$, Initial Prompts $\mathcal{P}^{(0)}$, Beam size $K$, Epochs $E$, Rounds $T$.
2: **Initialize:** Global prompts $\mathcal{P} \leftarrow \mathcal{P}^{(0)}$, Misalignment Buffer $\mathcal{B}_{mis} \leftarrow \emptyset$.
3: **Initialize:** Beam states $B_i \leftarrow \{(p_i^{(0)}, 0.0)\}$ for all optimizable agents $v_i \in \mathcal{V}$.
4: **for** epoch $e = 1$ **to** $E$ **do**
5:    **for** each agent $v_i$ in **TopologicalSort**($\mathcal{G}$) **do**
6:       {*Phase 1: Beam Refresh (Handling Non-Stationarity)*}
7:       **if** $e > 1$ **and** $B_i \neq \emptyset$ **then**
8:          Sample validation batch $\mathcal{B}_{val} \sim \mathcal{D}$.
9:          **for** each candidate $p \in B_i$ **do**
10:             {Re-evaluate score against current global best $\mathcal{P}_i$ under new contexts}
11:             $J(p) \leftarrow R(p, \mathcal{P}_i; \mathcal{B}_{val}) - 0.5$.
12:          **end for**
13:          Sort $B_i$ in descending order of $J(p)$. {Re-rank candidates}
14:          **if** $J(B_i[0]) > J(\mathcal{P}_i)$ **then**
15:             $\mathcal{P}_i \leftarrow B_i[0]$. {Update global anchor if ranking shifts}
16:          **end if**
17:       **end if**
18:       {*Phase 2: Misalignment-Guided Evolutionary Search*}
19:       **for** round $t = 1$ **to** $T$ **do**
20:          **Sampling:** Construct batch $\mathcal{B}_{iter}$ by mixing:
21:             1. Random samples from $\mathcal{D}$.
22:             2. **Hard Negatives** from $\mathcal{B}_{mis}$.
23:          **Trace Collection:** Collect traces $\mathcal{T}$ via Eq. (1) using $\mathcal{P}$ on $\mathcal{B}_{iter}$.
24:          Update $\mathcal{B}_{mis}$ with new failure cases found in traces.
25:          Initialize candidates $\mathcal{P}_{look} \leftarrow \emptyset$.
26:          **for** each parent $p \in B_i$ **do**
27:             **Proposal:** Generate offspring $\mathcal{P}' \sim \mathcal{M}_{opt}(p, \mathcal{T})$ targeting misalignment cases.
28:             **for** each child $p' \in \mathcal{P}'$ **do**
29:                **Joint Evaluation:** Calculate reward via Eq. (4):
30:                $s(p') \leftarrow \alpha \cdot R_{loc} + \beta \cdot R_{look} + \theta \cdot R_{glob}$.
31:                Update cumulative score: $J(p') \leftarrow J(p) + s(p')$.
32:                $\mathcal{P}_{look} \leftarrow \mathcal{P}_{look} \cup \{p'\}$.
33:             **end for**
34:          **end for**
35:          **Selection:** $B_i \leftarrow \text{Top-}K(B_i \cup \mathcal{P}_{look})$.
36:          **Global Synchronization:**
37:          **if** $\max_{p \in B_i} J(p) > J(\mathcal{P}_i)$ **then**
38:             $\mathcal{P}_i \leftarrow \operatorname{argmax}_{p \in B_i} J(p)$. {Broadcast best prompt to peers}
39:          **end if**
40:       **end for**
41:    **end for**
42: **end for**
43: **return** Optimized Configuration $\mathcal{P}$.

---

**Predictor Prompt (Math)**

```
Let's think step by step.  Show your final answer bracketed between <answer> and
</answer> tags.
{context}
Question:  {question}
Reasoning:
```

**Reflector Prompt (Math)**

```
Please review the solution above and criticize where it might be wrong.  Show your
final answer bracketed between <answer> and </answer>.
Question:  {question}
Solution:  {context}
Feedback:  indicating the reflection of the solution given the question.
Correct answer:
```

**Predictor Prompt (Reasoning)**

```
Let's think step by step.  Show your final option with one letter bracketed between
<answer> and </answer> tags.
{context}
Question:  {question}
Reasoning:
```

**Reflector Prompt (Reasoning)**

```
Please review the solution above and criticize where it might be wrong.  Show your
final option with one letter bracketed between <answer> and </answer>.
Question:  {question}
Solution:  {context}
Feedback:  indicating the reflection of the solution given the question.
Correct answer:
```

**Predictor Prompt (Code)**

```
You are an expert Python programmer.  Write clean, efficient, and well-documented
code.
Problem:  {question}
{context}
Provide a complete and correct code implementation in python.
Code:
```

**Reflector Prompt (Code)**

```
Review the following Python code and identify potential bugs or improvements.
Problem:  {question}
Code:  {context}
Provide an improved version in python at the end of your response.
Improved Code:
```

# E. Optimized Prompts

---

**Agent 1: Predictor 1 (Math-500)**

You are a creative and rigorous mathematical reasoner.  Your primary role is to
produce a clear, step-by-step analytical solution to the given mathematical problem.

**Instructions:**
Your response must be structured as a formal mathematical derivation.  Follow these
stages:

1.  **Deconstruction & Planning:**
* Deconstruct the problem into its essential components and identify the key
mathematical concepts involved.
* Outline your solution strategy.  Consider alternative approaches (e.g., algebraic,
geometric, trigonometric) and state your chosen path.

2.  **Step-by-Step Execution:**
* Execute your plan with precision.  Prioritize analytical and symbolic solutions over
numerical approximations.
* If you test specific values, use them only to form a hypothesis, which you must then
prove generally.
* Present each step clearly, showing all necessary calculations and explaining the
reasoning behind them.

3.  **Final Verification:**
* Briefly review your work to confirm the logic, accuracy, and completeness of your
solution.  Ensure the result fully addresses the original question.

**CRITICAL: Final Answer Format**
The final answer must be presented ONLY ONCE at the very end of your response.  It
must be enclosed in <answer> tags.  The content inside the tags must be the final,
simplified, and exact mathematical value, with no additional text or units.

{context}
Question:  {question}

**Solution:**

---

## Agent 2: Reflector 1 (Math-500)

You are the Reflector, a master of mathematical verification. Your role is to serve as the final, authoritative check on a proposed solution. You must adopt a mindset of professional skepticism, meticulously re-deriving the solution to either validate it or expose its flaws. Your output must be a rigorous, step-by-step critique and a definitive, correct answer.

**Your Task:**
Rigorously analyze the provided 'Context' (a proposed solution) for the given 'Problem'. Your goal is to independently verify the solution's reasoning and result.

**Output Structure:**
Your response must follow this mandatory four-part structure.

1. **Analysis of Proposed Method:**
Neutrally summarize the approach and key steps taken in the provided solution. **Do not pass judgment on its correctness in this section.**

2. **Independent Verification and Critique:**
Begin by solving the problem from first principles to establish a gold-standard solution. As you detail your step-by-step reasoning, critically compare it to the original solution's method. Pinpoint the exact location and nature of any errors (e.g., calculation mistake, logical fallacy, misinterpretation of the problem). If the original solution is correct, confirm its validity by showing your parallel reasoning.

3. **Final Verdict:**
Conclusively classify the original solution using **one** of the following verdicts. Your justification must align strictly with these definitions.
* **Correct:** The original solution's method, reasoning, and answer are all sound and well-justified.
* **Partially Correct / Incomplete:** The original solution has a valid approach but contains gaps. This includes missing steps, poor explanations, or unhandled edge cases, even if the final answer is correct. The answer is not rigorously supported by the provided reasoning.
* **Correct Answer, Flawed Reasoning:** The original solution's final answer is correct only by coincidence. The method used is logically unsound, contains mathematical fallacies, or relies on incorrect principles (e.g., two errors cancel each other out).
* **Incorrect:** The original solution contains significant errors in its reasoning or calculations, leading to a wrong answer.

4. **Final Answer:**
Provide the definitive correct answer derived from your independent verification.

**CRITICAL Formatting Requirements:**
- Your reasoning must be clear, structured, and easy to follow.
- The final answer MUST be enclosed in <answer> and </answer> tags.
- The content inside the <answer> tags must be the minimal, canonical mathematical answer ONLY. For example: <answer>42</answer> or <answer>11/2</answer>. Do not include any explanatory text, units, or variable names inside the tags.
- If your verified solution yields multiple distinct answers and the problem does not specify which to choose, list all correct answers separated by a comma. For example: <answer>6, -4</answer>.

**Problem:**
{question}

**Context:**
{context}

**Agent 3: Predictor 2 (Math-500)**

You are an expert mathematician, serving as the primary solver in a reasoning system.
Your goal is to provide a rigorous, step-by-step solution to the given mathematical
problem, derived from first principles.

**Your Task:**
Construct a detailed, self-contained, and logical derivation of the solution. Your
reasoning must be clear and easy to follow.

**Output Structure:**
1.  **Problem Analysis:** Begin by briefly restating the problem's objective and key
information.
2.  **Step-by-Step Derivation:** Present your solution as a series of numbered steps.
* Use markdown headers for each step in the format: `### Step X: [Descriptive Title]`.
* For example: `### Step 1: Define the variables`.
* Each step must clearly explain the mathematical principles, formulas, and
calculations used.
3.  **Verification:** If applicable, add a final step to check if your answer is
correct and satisfies the problem's conditions.

**CRITICAL FORMATTING RULES:**
* **Final Answer:** The final, simplified mathematical answer must appear **ONLY
ONCE**, at the very end of your response. It must be enclosed in <answer> and
</answer> tags.
* **Correct:** <answer>42</answer> or <answer>
frac{3}{4}</answer>
* **Incorrect:** <answer>The answer is 42</answer>, <answer>x = 42</answer>, or
<answer>
boxed{42}</answer>
* **Context:** The provided `context` is for high-level guidance only. **You MUST
derive the solution independently.** Do not copy or paraphrase reasoning from the
context. Your entire derivation must be your own work.

---
{context}

**Question:** {question}

**Solution:**

**Agent 4: Reflector 2 (Math-500)**

```
You are the Reflector, a critical auditor in a multi-agent mathematical reasoning
system.  Your role is to conduct a rigorous, structured audit of a proposed solution
to ensure its mathematical and logical integrity.  Your analysis must be meticulous,
constructive, and follow a precise five-step audit process.

You will conduct your audit in five steps:

1.  **Deconstruct the Problem:** Briefly restate the problem's core objective to
confirm your understanding of the goal.

2.  **Create Verification Checklist:** Distill the `Context` into a numbered list
of the specific, testable claims that form the core of the proposed solution's
argument.  Do not judge correctness yet.  This checklist will be your guide for the
audit.  Example:  "The solution's logic relies on these claims:  1.  The sum of three
consecutive integers can be written as 3n.  2.  The smallest cube divisible by 3 is 27.
3.  27 can be formed by 8+9+10."

3.  **Execute Verification:** Rigorously and independently verify each claim from
your Step 2 checklist.  Show your work clearly for each point, providing the necessary
calculations or logical steps.  This is your audit trail.  If the proposed logic is
flawed, continue your derivation to find the correct solution.

4.  **Deliver Verdict and Justification:** Compare the proposed logic against
your audit trail from Step 3 and issue a final verdict.  Justify your decision by
referencing the specific claims from your checklist.
* **Verdict:  Correct.** The proposed solution is mathematically sound and logically
complete.  All claims on the checklist were verified.  Briefly comment on the quality
of the original reasoning (e.g., "The method was efficient and direct").  If you found
a more elegant or insightful method, mention it as an alternative.
* **Verdict:  Incomplete Reasoning.** The final answer is correct, but the reasoning
is flawed or insufficient.  Pinpoint the specific claim that was unproven, assumed, or
represented a logical leap.  This verdict applies when the argument is not rigorous,
even if the conclusion is right.
* **Verdict:  Incorrect.** The proposed solution contains a mathematical error or a
fatal logical flaw.  Pinpoint the *first* claim on your checklist that was proven
false and explain the error.

5.  **Final Answer:** Conclude with the definitive correct answer from your
independent derivation.

CRITICAL FORMATTING INSTRUCTIONS:
- The final answer MUST appear ONLY ONCE at the very end of your response.
- The final answer MUST be enclosed in <answer> and </answer> tags.
- The tags MUST contain ONLY the canonical mathematical answer (e.g., a number, a
simplified fraction, an expression).  Do not include any explanatory text, units, or
reasoning within the tags.

---

### Problem
{question}

### Context
{context}

### Your Critical Audit
```

**Agent 1: Predictor 1 (GPQA)**

```
Analyze the problem to identify the key information and fundamental principles
required for the solution.  Develop a clear, step-by-step derivation where each step
logically follows from the data and the principles you identified.  Explain your
reasoning throughout the process to justify the final answer.
{context}
Question:  {question}
```

**Agent 2: Reflector 1 (GPQA)**

```
You are a critical reviewer verifying a proposed solution.

Critically evaluate the reasoning and calculations within the provided context.  Your
analysis should pinpoint specific flaws and provide the corrected logic.  If the
solution is sound, concisely confirm its key arguments instead of re-solving the
entire problem.

Conclude with the final answer, which must be a single letter enclosed in <answer>
tags.  Example:  <answer>A</answer>.
Question:  {question}
Context:  {context}
```

**Agent 3: Predictor 2 (GPQA)**

```
Critically analyze the question and context.  Construct a deductive, step-by-step
argument that logically progresses from the problem statement to the final answer.
Each step must be explicitly justified using the provided information and relevant
principles.  Systematically evaluate the options based on your argument.  Conclude
your entire output with the final answer formatted strictly as <answer>X</answer>,
containing only the single letter of the correct option.
Question:  {question}
Context:  {context}
```

**Agent 4: Reflector 2 (GPQA)**

```
You are a critical reviewer tasked with evaluating a proposed solution and determining
the correct answer.

Critically analyze the reasoning provided in the context.  Focus on identifying any
flaws in its logic, calculations, or assumptions.  If the context is flawed, concisely
explain the error and your corrected reasoning.  If the context is sound, you may
simply affirm its conclusion.

End your response with the single correct option letter enclosed in <answer> tags.
Question:  {question}
Context:  {context}
```

**Agent 1: Predictor 1 (HumanEval-ET)**

```
You are an expert Python programmer.  Your task is to provide a step-by-step thinking
process and then a final, correct Python function to solve the problem.

**Instructions:**
1.  **Analyze and Plan:**
* **Deconstruct the Goal:** Briefly state the primary objective.
* **Identify All Constraints & Definitions:** List every rule and definition.  Pay
close attention to how examples clarify terms (e.g., handling of spaces, what
constitutes a "word").
* **Propose an Algorithm:** Outline the step-by-step logic.  Consider potential
pitfalls or edge cases revealed by the examples.

2.  **Verify Logic Against Examples:**
* For *each* provided example, meticulously trace your proposed algorithm.  Show the
state of key variables at each step to prove your logic is sound.
* At the end of each trace, **explicitly compare your result with the example's
expected output.**
* If a mismatch occurs, **you MUST revise your algorithm and re-verify** before
proceeding.

3.  **Provide Final Code:**
* Write the final Python function.  **This code must be a direct and faithful
implementation of your verified algorithm.** Do not add or change logic at this stage.

**CRITICAL CODING RULES:**
* The function must implement only the pure algorithm.

**STRICTLY FORBIDDEN IN FINAL CODE:**
* Type hints (e.g., 'def func(arg:  str):')
* Docstrings or comments ('"""..."""' or '#')
* 'try...except' blocks
* Helper or nested functions
* Input validation or any conditional logic not explicitly required by the problem's
core logic.

Problem:  {question}
{context}
```

**Agent 2: Reflector 1 (HumanEval-ET)**

```
You are a Reflector agent, a senior code reviewer and debugger.  Your role is to
analyze a problem description and a 'context' describing a proposed solution.  Your
goal is to identify logical errors or inefficiencies in the proposal and then write a
corrected, highly-optimized Python function.

**Problem**:  {question}
**Context**:  {context}

First, provide a step-by-step analysis of the errors or inefficiencies in the solution
described by the context.  If the context's logic is sound, instead analyze potential
implementation pitfalls.  Conclude your analysis with a clear plan for the correct and
most efficient implementation.

Then, generate the corrected Python code, following these absolute rules.

**CRITICAL CODE GENERATION RULES:**
Your final output must be a single markdown block containing **only** the Python
function.
* **Signature:** The function signature must be 'def function_name(arguments):'.  Do
**NOT** use type hints.
* **Body:** Implement the most direct and computationally efficient logic possible.
Do **NOT** include 'import' statements, comments, helper functions, or 'try/except'
blocks.
* **Assumptions:** Assume inputs are always valid as described in the problem.  Do
**NOT** add input validation or unnecessary edge-case handling.

Example of a perfect response:
```python
def function_name(input_arguments):
# The most direct and efficient logic
return output
```
```

**Agent 3: Predictor 2 (HumanEval-ET)**

```
You are an expert Python programmer providing a direct solution to a coding problem.
Your entire response must strictly follow this two-part structure.

*Part 1:  Reasoning Plan**
Provide a clear, numbered, step-by-step plan.
1.  **Analyze Goal:** State the function's main objective.
2.  **Analyze Examples:** Critically examine the provided examples.  If an example
contradicts the problem description, state the contradiction and confirm your
implementation will follow the example's logic.
3.  **Algorithm:** Outline the most concise and efficient step-by-step algorithm
to solve the problem.  Mention iteration direction (e.g., forward, reverse) if
applicable.
4.  **Edge Cases:** Briefly describe how the algorithm handles key boundary conditions
(e.g., n=1, empty inputs).

**Part 2:  Python Code**
Provide the final, complete Python function inside a single code block.

**CODE BLOCK RULES (MANDATORY):**
1.  **NO TYPE HINTS:** The function signature must be 'def function_name(arg1, arg2):'.
You MUST remove any type hints (e.g., 'arg1:  str') found in the problem description.
2.  **NO COMMENTS OR DOCSTRINGS.**
3.  **NO 'try-except' BLOCKS.**
4.  **CORE LOGIC ONLY:** Implement only the essential logic.  Do not add input
validation (e.g., type checking) or handle cases not explicitly covered by the
problem.

Problem:  {question}
{context}
```

---

**Agent 4: Reflector 2 (HumanEval-ET)**

```
You are a meticulous code reviewer.  Your purpose is to analyze a flawed Python
solution described in the context, explain its specific errors with clear reasoning,
and provide a perfectly corrected version.

Problem:  {question}
Context (Description of a flawed solution):  {context}

First, write a step-by-step **Analysis of Errors**.
Your analysis must be based *only* on the solution described in the 'Context'.
Pinpoint its specific logical flaws by referencing the **exact requirements** and
examples from the problem description.  **Crucially, if the problem description is
ambiguous, the provided examples are the absolute source of truth for the expected
behavior.**

Next, provide the **Corrected Code**.
Your code must fix the specific errors identified in your analysis.  Before writing
the code, you must mentally walk through your new logic with **every single example**
from the problem description to guarantee it is flawless.  A failure in any example is
a failure of the entire task.

Strictly adhere to the following rules for the code block:
1.  Provide **only** the function definition, starting with 'def'.
2.  The function signature **MUST NOT** contain any type hints, docstrings, or
comments.
3.  The function must be self-contained.  **DO NOT** define helper functions.
4.  **DO NOT** include 'import' statements, test cases, or 'print' statements.
5.  Implement only the core algorithm.  Do not add input validation or exception
handling.

**Analysis of Errors:**
<Your step-by-step reasoning based on the Context here>

**Corrected Code:**
```python
# Your self-contained, verified Python code here
```
```

## F. Detailed Sensitivity Analysis of Joint Evaluation Weights

In this section, we provide the detailed numerical results corresponding to the sensitivity analysis discussed in Section 4.3 (Impact of Joint Evaluation Strategy). Table 4 enumerates the performance scores obtained under a grid search of hyperparameter configurations for the three evaluation dimensions: local adherence ($\alpha$), lookahead potential ($\beta$), and global alignment ($\theta$). As evidenced by the empirical results, the configuration with $\alpha = 4, \beta = 4, \theta = 2$ (corresponding to normalized weights of $0.4, 0.4, 0.2$) emerges as the optimal setting, achieving the highest average performance of 70.39 across all benchmarks. This superior performance yields two critical insights into MAS optimization:

1. **Dominance of Intermediate Signals:** The balanced heavy weighting on Local Validity ($\alpha$) and Lookahead Potential ($\beta$) outperforms configurations skewed towards Global Alignment ($\theta$). This validates our hypothesis that in multi-step reasoning chains, the final outcome provides a supervision signal that is too sparse and noisy for intermediate agents. In contrast, $\alpha$ and $\beta$ offer dense, immediate feedback.

2. **Synergy of Correctness and Utility:** The equal importance of $\alpha = 4$ and $\beta = 4$ suggests that for an agent to be effective, it is not enough to merely be "locally correct" (satisfying self-consistency); it must also be "topologically useful" (facilitating the success of its successor). The lower weight on $\theta = 2$ serves as a necessary but auxiliary weak constraint to ensure the overall trajectory does not drift from the final goal.

*Table 4.* Performance comparison under different weight configurations for joint evaluation. The weights $\alpha$, $\beta$, and $\theta$ correspond to local adherence, lookahead potential, and global alignment, respectively. The results confirm that prioritizing local and lookahead signals (e.g., 4:4:2 configuration) generally outperforms configurations heavily weighted towards global alignment.

| Evaluation Weights | | | Task Performance | | | | | | |
|---|---|---|---|---|---|---|---|---|---|
| $\alpha$ | $\beta$ | $\theta$ | MATH-500 | AGIEval | AQuA | GPQA | MBPP | HumanEval | Avg. |
| 2 | 3 | 5 | 77.80 | 63.28 | 87.80 | 50.51 | 64.42 | 72.93 | 69.46 |
| 2 | 4 | 4 | 78.00 | 62.11 | 85.04 | 52.53 | 66.51 | 75.61 | 69.97 |
| 2 | 5 | 3 | 75.60 | 62.50 | 86.85 | 50.51 | 63.23 | 73.17 | 68.64 |
| 3 | 2 | 5 | 77.00 | 63.28 | 85.04 | 53.03 | 64.42 | 71.05 | 68.97 |
| 3 | 3 | 4 | 78.40 | 64.06 | 84.68 | 50.00 | 63.70 | 72.56 | 68.90 |
| 3 | 4 | 3 | 75.60 | 65.62 | 86.22 | 54.55 | 63.29 | 71.95 | 69.54 |
| 4 | 2 | 4 | 76.80 | 59.38 | 84.65 | 54.04 | 65.81 | 70.12 | 68.47 |
| 4 | 3 | 3 | 79.40 | 61.72 | 85.83 | 51.01 | 66.04 | 74.39 | 69.73 |
| 4 | 4 | 2 | 77.80 | 61.98 | 85.56 | 58.08 | 65.11 | 73.78 | 70.39 |
| 5 | 2 | 3 | 77.40 | 59.77 | 84.89 | 51.01 | 63.23 | 73.17 | 68.25 |
| 5 | 3 | 2 | 77.00 | 65.23 | 87.01 | 52.53 | 65.81 | 72.56 | 70.02 |

## G. Integration with Topology Optimization Frameworks

To assess the generalizability of our framework, we investigate the compatibility of MASPO with existing methods focused on topology optimization. Specifically, we apply MASPO on top of AgentDropout, a baseline that optimizes the communication topology by selectively pruning agent interactions. As presented in Table 5, the integration of MASPO yields consistent performance enhancements over the standard AgentDropout baseline. This result highlights two critical insights that while AgentDropout optimizes the topology of MAS, MASPO refines the *instructions* within the agent (node). The observed improvements demonstrate that these two optimization dimensions: topological structure and prompt semantics are orthogonal and can be combined to achieve additive gains. The efficacy of MASPO is not confined to standard sequential or hierarchical graphs. It adapts effectively to the specialized, dynamic topologies induced by structural optimization frameworks, confirming that our joint evaluation and misalignment mining mechanisms remain robust across diverse architectural configurations.

*Table 5.* The performance compared with integrating MASPO with the structural optimization framework AgentDropout.

| Method | MATH-500 | AGIEval-MATH | AQuA | GPQA | MBPP | HumanEval-ET | Avg. |
|---|---|---|---|---|---|---|---|
| AgentDropout | 76.80 | 59.77 | 86.23 | 47.98 | 58.09 | 72.44 | 66.89 |
| + MASPO | 78.20 | 62.98 | 86.61 | 54.04 | 66.51 | 74.39 | 70.46 |

## H. Impact of Sample Pool Size

To investigate the influence of the sample pool size on the optimization efficacy of MASPO, we conduct comparative experiments with varying configurations of $|\mathcal{D}|$. As presented in Table 6, increasing the pool size yields consistent improvements. However, when the sample pool size exceeds 50, the marginal performance gains become negligible. This saturation phenomenon suggests that a pool of 50 samples provides sufficient diversity for effective mini-batch sampling during the iterative optimization process. Beyond this threshold, additional samples contribute diminishing returns, likely because the existing pool already captures the representative distribution of task instances required for robust prompt evolution and evaluation. Based on these empirical findings, we adopt $|\mathcal{D}| = 50$ as the default configuration, which achieves an optimal balance between optimization effectiveness and computational efficiency.

*Table 6.* The performance of MASPO under different sample pool sizes $|\mathcal{D}|$.

| Size of $\mathcal{D}$ | MATH-500 | AGIEval-MATH | AQuA | GPQA | MBPP | HumanEval-ET | Avg. |
|---|---|---|---|---|---|---|---|
| 30 | 77.00 | 60.13 | 85.83 | 54.04 | 64.64 | 73.17 | 69.13 |
| 50 (default) | 77.80 | 61.98 | 85.56 | 58.08 | 65.11 | 73.78 | 70.39 |
| 70 | 77.60 | 62.11 | 86.22 | 56.06 | 65.57 | 75.00 | 70.43 |

