# OpenReview forum: "MASPO: Joint Prompt Optimization for LLM-based Multi-Agent Systems"
_ICML.cc/2026/Conference — ICML 2026 regular_

### Official Review · Reviewer_VApu · 2026-02-25

**Soundness:** 3
**Presentation:** 2
**Significance:** 3
**Originality:** 3
**Overall Recommendation:** 4
**Confidence:** 4

**Summary:**

1. This paper proposes MASPO, a framework for jointly optimizing prompts within LLM-based MAS frameworks. The method consists of three main components:
   - A multi-granularity reward modeling mechanism, integrating local validity, lookahead potential, and global alignment.
   - A misalignment-aware prompt rewriting pipeline, which mines coordination failure cases to guide optimization.
   - A tree-based evolutionary beam search algorithm with a coordinate-ascent-style scheduling protocol and beam refresh mechanism to mitigate non-stationarity.
2. This paper evaluates MASPO on multiple benchmarks. Experimental results show that the proposed method improves MAS performance without relying on ground-truth labels and that optimized prompts can transfer across different backbone models. Ablation studies are conducted to verify the contribution of each component.

**Compliance With Llm Reviewing Policy:**

Affirmed.

**Final Justification:**

Most of my initial concerns have been satisfactorily resolved. The remaining issue relates to the clarity of writing, which the authors have acknowledged and committed to improving in the revised version. Therefore, I have raised my score accordingly.

**Key Questions For Authors:**

1. **Multi-step lookahead.** The reward modeling considers only one-step lookahead. Why not incorporate deeper lookahead for downstream agents? Have you tested multi-hop evaluation?
2. **Negative reward handling.** Is it reasonable to assume non-negative rewards? How does the framework handle the case where all the rewritten prompts are worse than the original one?
3. **Computational complexity.** Can you provide a formal complexity analysis in terms of number of inference calls, with respect to the number of agents, beam width, and optimization rounds?
4. **Optimization cost.** What is the average optimization time and inference budget compared to the baselines?
5. **Model choice.** Why is Qwen3-8B used as the main backbone instead of frontier SOTA models such as GPT-5 or Gemini-3?
6. **MAS structure clarification.** Can you explain what "sequential MAS" and "hierarchical MAS" mean? Why not test on strong paradigms such as multi-agent debate?
7. **Scalability of misalignment sampling.** Why does increasing misalignment injection not monotonically improve performance? Is there an intuitive explanation?

**Limitations:**

The limitations and potential negative societal impact have been discussed in this paper.

**Strengths And Weaknesses:**

### Strengths

1. **Novel and important research question.** While prompt optimization has been widely studied in single-agent settings, the problem of jointly optimizing prompts in MAS is timely and underexplored. Addressing related challenges is critical for scaling collaborative LLM systems.
2. **Systematic and principled framework.** The proposed method is well-structured and conceptually coherent. The overall design of the optimization process is reasonable and modular, and in principle can be applied to various MAS architectures.
3. **Strong empirical validation.** The experiments are relatively comprehensive. Multiple benchmarks and MAS structures are evaluated. Cross-model transfer experiments and detailed ablation studies are conducted. The reported gains are consistent and reproducible with released code, which is commendable.

### Weaknesses

1. **Reliability of one-step lookahead evaluation.** The reward modeling evaluates lookahead potential based on immediate successor outputs. However, one-step lookahead may not sufficiently approximate long-horizon effects in deeper MAS graphs. This raises concerns about whether the proposed reward truly captures global causal impact.
2. **Assumption of non-negative reward for rewritten prompts.** The optimization framework seems to implicitly assume rewritten prompts are improvements. However, a prompt rewrite can easily degrade performance. If reward signals are always non-negative or relative to a baseline without penalizing deterioration strongly enough, the search may drift.
3. **Optimization cost not reported.** Each candidate evaluation requires executing the entire MAS pipeline. This implies high inference cost and large number of forward passes. However, the paper does not provide computational complexity analysis, number of inference calls, and wall-clock time comparison.
4. **Limited backbone diversity in main experiments.** The main experiments use Qwen3-8B as the backbone and do not include frontier SOTA models (e.g., GPT-5, Gemini-3-level models). If the method is claimed to be generally applicable, validation on stronger models should be included in core experiments rather than only in transfer evaluation.
5. **MAS structures underspecified.** The definitions of "sequential MAS" and "hierarchical MAS" are not sufficiently detailed in the main paper. The architectural differences and communication topology should be more clearly explained.
6. **Scalability concerns from misalignment sampling.** The ablation shows performance peaks at a moderate misalignment injection size. Increasing injection volume does not continually improve results. This suggests diminishing returns, limited scalability, and possible noise amplification.
7. **Writing and organization issues.** Several writing issues reduce the clarity of this paper: (1) The discussion of prior works (Line 22) should be clearly separated into distinct paragraphs for better readability. (2) The main contribution (Line 65) should be stated once in a focused manner without unnecessary repetition. (3) Presenting misalignment sampling (Line 155) before introducing reward modeling causes confusion. Placing the reward modeling section before trajectory generation would result in a more logical flow.

---

> ### Author Rebuttal · Authors · 2026-03-30
>
> > **Weakness 1 & Question 1:** One-step lookahead reliability
>
> Thank you for this insightful question. In our joint reward model (Eq. 5), one-step Lookahead Potential works synergistically with Global Alignment, which evaluates $o_{\text{glob}}$ and thus implicitly captures multi-hop dependencies across the entire agent chain. To further validate this design, we conducted additional experiments with multi-step lookahead:
> | Method| MATH-500 | AGIEval-MATH | AQuA  | GPQA  | MBPP  | HumanEval-ET | Avg.  |
> |-|-|-|-|-|-|-|-|
> | Sequential MAS | 75.10 | 59.38 | 83.47 | 47.73 | 57.26 | 68.90 | 65.31 |
> | SPO | 77.20    | 60.13    | 81.10 | 49.52 | 63.47 | 67.94    | 66.56 |
> | MASPO | 77.80    | 61.98  | 85.56 | 58.08 | 65.11 | 73.78     | 70.39 |
> | +2-step    | 78.00    | 62.33 | 85.86 | 57.58 | 64.04 | 74.73   | 70.42 |
> | +3-step   | 77.80    | 62.65  | 84.85 | 57.07 | 65.11 | 75.00  | 70.41 |
>
> The results show that deeper lookahead yields negligible improvement, confirming that Global Alignment already captures long-range effects. We adopt one-step lookahead for its favorable efficiency-performance trade-off.
>
> > **Weakness 2 & Question 2:** Non-negative reward assumption
>
> Thank you for raising this point. In MASPO, rewards are compared relatively across candidates, Beam Search with Top-K selection inherently filters degraded candidates. The Beam Refresh mechanism (Eq. 8) further centers scores by subtracting 0.5, explicitly assigning negative scores to candidates worse than the current best for elimination. Thus, even with non-negative raw rewards, inferior prompts are effectively excluded. To further verify, we conducted an experiment with explicit negative rewards (R-0.5) and included parent nodes in Top-K selection:
> | Method | MATH-500 | AGIEval-MATH | AQuA  | GPQA  | MBPP  | HumanEval-ET | Avg.  |
> |-|-|-|-|-|-|-|-|
> | MASPO  | 77.80 | 61.98  | 85.56 | 58.08 | 65.11 | 73.78 | 70.39 |
> | +negative reward | 77.20| 61.67| 84.85 | 56.57 | 64.64 | 72.56 | 69.58 |
>
> The results confirm that MASPO's non-negative reward design is both reasonable and effective.
>
> > **Weakness 3 & Questions 3, 4:** Optimization cost and complexity
>
> Thank you for this important concern. We provide the formal complexity analysis below. Evaluating one candidate requires $\sim|B| \times N_{\text{downstream}}$ inference calls plus $\sim|B| \times (1 + |N_{\text{out}}| + 1)$ evaluator calls. With $K \times K_{\text{sub}}$ candidates per step, the total complexity is:
> $C_{\text{total}} = O(D \cdot T \cdot K \cdot K_{\text{sub}} \cdot |B| \cdot N^2).$
>
> Please refer to our response to Reviewer p8hr (Weakness 4 & 5) for detailed cost comparison and amortization analysis.
>
> > **Weakness 4 & Question 5:** Backbone model diversity
>
> Thank you for this question. We selected Qwen3-8B for two reasons:
>
> (1)MAS involve multi-agent multi-round interactions compounded by iterative optimization, with costs far exceeding single-agent settings, recent MAS works [1][2] similarly adopt small-scale LLMs;
>
> (2)Table 3 shows MASPO-optimized prompts transfer effectively to stronger models (Gemini-2.5-Pro, Claude-Sonnet-4, etc.) with consistent gains, demonstrating that MASPO captures generalizable interaction logic rather than model-specific artifacts.
>
> [1] Stronger-MAS: Multi-Agent Reinforcement Learning for Collaborative LLMs. ICLR 2026.
>
> [2] Thought Communication in Multiagent Collaboration. NeurIPS 2025
>
>
> > **Weakness 5 & Question 6:** MAS structure clarification
>
> Thank you for the concern. Our paper does describe both architectures, but we provide further clarification here. Appendix D details the Sequential MAS's chain topology (Predictor→Reflector→Predictor→Reflector) and its iterative refinement mechanism. The Hierarchical MAS strictly follows the two-layer parallel-aggregation architecture of Zou et al. (2025). Complete initial prompts for all agents are also provided in Appendix D.
>
> Regarding multi-agent debate, recent research [1] shows its gains primarily stem from voting (analogous to self-consistency in Table 1) rather than debate itself. The Hierarchical MAS serves as a stronger baseline.
>
> [1] Debate or Vote: Which Yields Better Decisions in Multi-Agent Large Language Models? NeurIPS 2025.
>
> > **Weakness 6 & Question 7:** Misalignment sampling scalability
>
> Thank you for the careful observation. Misalignment cases serve as "hard negatives" that provide high-information gradient signals, helping the optimizer distinguish prompts that are locally effective but globally harmful. However, when the injection volume is excessive, the over-representation of misalignment cases in the optimization batch causes distribution shift and noise amplification. As shown in Table 2, performance peaks at $K_{\text{mis}}=3$, achieving the optimal balance between diagnostic signal and distributional representativeness.
>
> > **Weakness 7:** Writing and organization
>
> Thank you for these constructive suggestions. We fully agree with all three points and will revise accordingly.

---

> > ### Author Rebuttal · Reviewer_VApu · 2026-04-03
> >
> > Thank you to the authors for their thoughtful and detailed responses. I appreciate the effort made to address my concerns during the rebuttal phase.
> >
> > Most of my initial concerns have been satisfactorily resolved. The remaining issue relates to the clarity of writing, which the authors have acknowledged and committed to improving in the revised version.
> >
> > In light of these updates, I have revised my score and confidence accordingly.

---

### Official Review · Reviewer_cL4u · 2026-03-12

**Soundness:** 3
**Presentation:** 4
**Significance:** 3
**Originality:** 3
**Overall Recommendation:** 4
**Confidence:** 4

**Summary:**

This paper introduces MASPO, a joint prompt optimization framework for LLM-based multi-agent systems, specifically targeting the critical Local-Global Misalignment between local agent objectives and holistic system goals. The key contribution is a Multi-Granularity Joint Evaluation and Reward Mechanism that incorporates Local Validity, Lookahead Potential, and Global Alignment to effectively resolve the complex credit assignment problem across interacting agents. To explicitly bridge the local-global gap, the framework employs a memory buffer to mine and inject Misalignment Cases, specific scenarios where agents succeed locally but induce system-wide failures, thereby guiding the optimizer. Furthermore, the design utilizes a coordinate ascent-style optimization protocol augmented with a Beam Refresh Mechanism, ensuring stable co-adaptation between upstream and downstream agents in a non-stationary environment. The methodology is rigorously developed, and comprehensive experimental evaluations across diverse datasets, alongside thorough ablation studies, demonstrate the effectiveness of both the overall framework and its individual core components.

**Compliance With Llm Reviewing Policy:**

Affirmed.

**Final Justification:**

I believe prompts are of significant importance to problem-solving in multi-agent systems. The paper's proposed joint prompt optimization method provides an exploration and reference for this research direction, with a solid overall presentation. The authors' rebuttal has addressed my prior concerns and reinforced my prior assessment. I therefore maintain my recommendation of weak accept for this paper.

**Key Questions For Authors:**

1. The paper introduces a misalignment buffer to store failure cases and reuse them during subsequent optimization. However, the specific maintenance strategy for this buffer warrants further elaboration. For example: Do misalignment cases accumulate continuously across iterations, or is the buffer cleared or refreshed at certain stages? If cases accumulate, do early cases continue to influence the optimizer? How are a fixed number of failure cases selected or sampled from this buffer during the optimization process? Providing further clarification on how the misalignment buffer is updated and managed throughout the optimization process would improve the reproducibility and understanding of the proposed method.

2. The proposed framework achieves consistently promising experimental results across various tasks. Each agent is initialized with a specific initial prompt at the start of the process. However, in practical applications, users are likely to provide low-quality instructions. Can the framework still achieve solid task performance under such poor initialization conditions?

**Limitations:**

The paper includes an "Impact Statement" that briefly discusses potential societal implications, but it does not adequately address the limitations of the proposed method.
The experimental evaluation is conducted on a limited set of multi-agent coordination settings and tasks. Given the large number of existing multi-agent frameworks and evaluation benchmarks, it would be valuable to further validate the proposed method across a broader range of agent architectures and task environments to better demonstrate its generality.

**Strengths And Weaknesses:**

Strengths
The Multi-Granularity Joint Evaluation and Misalignment-Aware Sampling method used in the paper can effectively measure the quality of prompts and promote the generation of locally and globally effective prompts. The overall structure of the paper is clear, and the elaboration of the problem, the introduction of the method, and the setting of the experiments are all relatively clearly written.

Weaknesses
Based on Figure 3, the claim that "MASPO consistently reduces the misalignment rate as optimization depth increases, and remains lower than other variants" appears overly absolute.

---

> ### Author Rebuttal · Authors · 2026-03-30
>
> Thank you for the review.
> > **Weakness 1:** Overly absolute claim on misalignment rate reduction
>
> Thank you for your detailed questions regarding the description of Figure 3. We agree with your observation. The misalignment rate does exhibit minor fluctuations between individual optimization steps. We will revise the claim to: "MASPO exhibits a clear overall downward trend in misalignment rate as optimization depth increases, and generally maintains a lower misalignment rate compared to other variants across all three task domains."
>
> > **Question 1:** Misalignment buffer maintenance strategy
>
> Thank you for your in-depth questions regarding the misalignment buffer management strategy. **The misalignment buffer does not accumulate indefinitely.** Specifically, the buffer is re-collected and overwritten at each agent's optimization round based on the current best prompt, rather than carried over across rounds. This ensures that early misalignment cases do not persistently influence subsequent optimization, keeping the feedback signal synchronized with the current optimization state. This design is motivated by the non-stationary nature of MAS optimization: as upstream prompts evolve, previously identified misalignment patterns may no longer be relevant, and stale cases could mislead the optimizer. Regarding sampling, as described in Section 4.1, we fix the maximum number of sampled misalignment cases at $K_{mis}=3$ per optimization batch, which are injected alongside regular samples to guide the prompt rewriting process.
>
> > **Question 2:** Robustness to poor prompt initialization
>
> Thank you for raising the issue regarding the initial prompt. To verify MASPO's effectiveness under poor initialization conditions, we designed two experiments: (1) **Minimal initialization**, where all agent prompts are replaced with a minimal instruction "Answer the question:" containing no task-specific guidance; (2) **Wrong-domain initialization**, where prompts are cross-assigned to mismatched domains (math prompts → code tasks, code prompts → reasoning tasks, reasoning prompts → math tasks), simulating a scenario where users provide entirely irrelevant instructions.
> | Method              | MATH-500 | AGIEval-MATH | AQuA  | GPQA  | MBPP  | HumanEval-ET | Avg.  |
> |---------------------|----------|--------------|-------|-------|-------|--------------|-------|
> | Sequential MAS      | 75.10    | 59.38        | 83.47 | 47.73 | 57.26 | 68.90        | 65.31 |
> | TPE                 | 75.80    | 58.13        | 84.92 | 48.04 | 61.30 | 70.12        | 66.49 |
> | SPO                 | 77.20    | 60.13        | 81.10 | 49.52 | 63.47 | 67.94        | 66.56 |
> | MASPO               | 77.80    | 61.98        | 85.56 | 58.08 | 65.11 | 73.78        | 70.39 |
> | + Minimal Init.     | 77.20    | 62.23        | 86.61 | 56.06 | 64.64 | 72.95        | 69.95 |
> | + Wrong-domain Init.| 77.00    | 61.62        | 85.86 | 55.56 | 65.11 | 73.17        | 69.72 |
>
> Both degraded initialization settings show only marginal performance drops compared to the default MASPO, while still substantially outperforming all baselines. This demonstrates that MASPO can effectively recover from low-quality initializations and is robust to the quality of user-provided initial prompts.

---

> > ### Author Rebuttal · Reviewer_cL4u · 2026-04-03
> >
> > Thank the authors for the response and the additional experiments. I will maintain my original score.

---

### Official Review · Reviewer_YbbX · 2026-03-15

**Soundness:** 3
**Presentation:** 3
**Significance:** 3
**Originality:** 3
**Overall Recommendation:** 4
**Confidence:** 4

**Summary:**

This paper studies joint prompt optimization for LLM-based multi-agent systems. The main claim is that optimizing prompts independently is insufficient because local agent quality may be misaligned with downstream or system-level success. To address this, the paper proposes MASPO, which combines three ingredients: (i) a joint reward with local validity, lookahead potential, and global alignment, (ii) misalignment-aware sampling that reuses failure cases where local success still harms the overall system, and (iii) a coordinate-ascent / beam-search procedure with beam refresh to mitigate non-stationarity across agents. Experiments on six benchmarks spanning math, reasoning, and code generation show improvements over vanilla prompting, several single-agent baselines, and prompt optimization baselines applied to sequential and hierarchical MAS

**Compliance With Llm Reviewing Policy:**

Affirmed.

**Final Justification:**

I believe the paper meets the bar for acceptance, and I do not see significant issues regarding its novelty.

**Key Questions For Authors:**

N/A

**Limitations:**

yes

**Strengths And Weaknesses:**

Strengths
- The paper tackles an important and timely problem. Prompt optimization for MAS is genuinely harder than single-agent prompt tuning because of inter-agent coupling, covariate shift, and credit assignment, and the paper articulates this motivation clearly.
- The method is reasonably well structured. The decomposition into joint evaluation, misalignment mining, and beam refresh is intuitive, and Figure 1 helps explain the workflow clearly. The formulation in Eq. (5) also makes the intended optimization target understandable
- The experimental section is fairly broad. The method is evaluated on six benchmarks and two MAS topologies, with several ablations on search, scheduling, and misalignment sampling. The cross-model transfer analysis is also a useful addition.

Weaknesses:
- The paper’s central evaluation mechanism remains fragile because it is entirely LLM-judged. Local validity, lookahead potential, and global alignment are all defined through pairwise preferences from the evaluator model. This makes the optimization target highly dependent on the evaluator’s biases, yet there is no human validation, no inter-evaluator agreement analysis, and no calibration study showing that the judge is reliable for intermediate-agent outputs. Since the method optimizes prompts against these judgments, evaluator bias can be directly amplified.
- The paper does not separate the gains from “joint reward design” versus “using a stronger optimizer/evaluator” versus “having a generative prompt mutator.” A stronger set of controls would include equal-budget comparisons, prompt-search budget comparisons, and variants where the same Gemini budget is used for simpler optimization heuristics. As written, it is hard to know whether MASPO is better because of its algorithmic ideas or because it allocates more powerful inference to optimization.
- The paper lacks sufficient qualitative analysis. Since the core claim is about repairing interaction breakdowns between agents, I expected concrete before/after examples showing how an optimized upstream prompt changes downstream reasoning trajectories, or examples where local-valid but globally harmful outputs are corrected. The appendix seems to contain prompt templates, but the main paper does not give enough qualitative evidence of what kinds of prompt edits MASPO actually discovers.

---

> ### Author Rebuttal · Authors · 2026-03-30
>
> Thank you for the review.
> > **Weakness 1**: LLM-judged evaluation reliability
>
> Thank you for your in-depth attention to the reliability of the evaluation mechanism. Our framework incorporates multiple robustness mechanisms to mitigate evaluator bias: the multi-granularity evaluation (Eq. 5) assesses each candidate from three independent dimensions, serving as implicit multi-annotator cross-validation; cumulative scoring (Eq. 7) and Beam Refresh (Eq. 8) further smooth single-judgment noise and prevent historical bias accumulation. As shown in Table 2 (IV), using the weaker Qwen3-8B as evaluator still yields consistent improvements, indicating the framework is not highly dependent on a specific evaluator's biases.
> To directly address the concern about human validation, we randomly sampled 50 pairwise judgments on MATH-500 and compared LLM evaluations against human annotations:
> | Evaluation Dimension  | Agreement Rate |
> |-----------------------|----------------|
> | Local Validity        |         78%  |
> | Lookahead Potential   |          76%  |
> | Global Alignment      |          86%  |
>
> The results confirm **a high level of agreement between LLM and human judgments**, supporting the reliability of our evaluation mechanism.
>
> > **Weakness 2:** Insufficient attribution of gains
>
> We appreciate this suggestion for more rigorous controls. The existing component ablation (Table 2, Part II) already demonstrates the effectiveness of each component under the same Gemini budget. To further isolate the contribution of algorithmic design, we conducted additional experiments where SPO is given equivalent computational resources: SPO + same search budget increases SPO's total number of optimization steps to match MASPO's, while SPO + same Gemini budget ensures SPO consumes the same total number of Gemini API calls as MASPO:
> | Method                 | MATH-500 | AGIEval-MATH | AQuA  | GPQA  | MBPP  | HumanEval-ET | Avg.  |
> |------------------------|----------|--------------|-------|-------|-------|--------------|-------|
> | Sequential MAS         | 75.10    | 59.38        | 83.47 | 47.73 | 57.26 | 68.90        | 65.31 |
> | TPE                    | 75.80    | 58.13        | 84.92 | 48.04 | 61.30 | 70.12        | 66.49 |
> | SPO                    | 77.20    | 60.13        | 81.10 | 49.52 | 63.47 | 67.94        | 66.56 |
> | SPO + same search      | 76.80    | 61.33        | 84.85 | 50.51 | 63.70 | 69.52        | 67.79 |
> | SPO + same Gemini      | 77.60    | 60.67        | 85.04 | 51.01 | 63.93 | 68.90        | 67.86 |
> | MASPO                  | 77.80    | 61.98        | 85.56 | 58.08 | 65.11 | 73.78        | 70.39 |
>
> Even with equivalent computational resources, SPO substantially underperforms MASPO, confirming that the performance gains are **primarily attributable to MASPO's algorithmic design rather than resource allocation**.
>
> > **Weakness 3:** Lack of qualitative analysis
>
> We appreciate this suggestion for sufficient qualitative analysis. We agree that more concrete examples would strengthen the paper. Below is a representative before/after case showing how MASPO optimizes the Reflector agent's prompt:
>
> **Before optimization**:
>
> "Please review the solution above and criticize where it might be wrong..."
>
> **After optimization**:
>
> "You are the Reflector, a master of mathematical verification. You must adopt a mindset of professional skepticism, meticulously re-deriving the solution to either validate it or expose its flaws.
> Your response must follow this mandatory four-part structure:
> 1. Analysis of Proposed Method: Neutrally summarize the approach. Do not pass judgment on its correctness in this section.
> 2. Independent Verification and Critique: Solve the problem from first principles. Pinpoint the exact location and nature of any errors.
> 3. Final Verdict: Classify the original solution as: Correct / Partially Correct / Correct Answer, Flawed Reasoning / Incorrect
> 4. Final Answer..."
>
> MASPO automatically discovered three key modifications through iterative optimization: requiring independent verification ("solve from first principles"), forcing the Reflector to provide correct derivations rather than merely identifying issues; adding structured verdict categories, enabling downstream agents to distinguish subtle cases such as "correct answer but flawed reasoning"; and enforcing separation of analysis and judgment ("do not pass judgment in this section"), preventing premature conclusions that bypass rigorous verification.

---

> > ### Author Rebuttal · Reviewer_YbbX · 2026-04-03
> >
> > I thank the authors for their response. I will maintain my original score.

---

### Official Review · Reviewer_p8hr · 2026-03-16

**Soundness:** 2
**Presentation:** 2
**Significance:** 2
**Originality:** 2
**Overall Recommendation:** 3
**Confidence:** 4

**Summary:**

This paper proposes MASPO, a framework for joint prompt optimization in multi-agent LLM systems. The method aims to address the local–global misalignment problem in multi-agent pipelines, where prompts optimized for individual agents may harm downstream agents and degrade the final system output. MASPO introduces a joint evaluation signal combining local validity, downstream potential, and global alignment, and uses a trace-guided evolutionary beam search to optimize prompts across agents. Experiments on multiple reasoning and coding benchmarks show improvements over several prompt optimization baselines and demonstrate some transferability of optimized prompts across models.

**Compliance With Llm Reviewing Policy:**

Affirmed.

**Final Justification:**

Post rebuttal: The rebuttal effectively clarifies my concerns about performance comparison and cost. I still have reservations regarding novelty and the experimental setup. I increased my score from 2 to 3.

**Key Questions For Authors:**

See weakness

**Limitations:**

yes

**Strengths And Weaknesses:**

Strengths
- The idea of jointly evaluating prompts using local and downstream signals is well motivated and provides a reasonable approach to mitigating local–global misalignment.
- The empirical section is fairly broad, including multiple datasets, ablations, and transfer experiments.
- The transferability results, where optimized prompts generalize to other models, are interesting and suggest the learned prompts capture some reusable coordination patterns.

Weaknesses
- The overall novelty appears limited. The framework mainly combines existing techniques such as evolutionary prompt search, beam search, and LLM-based evaluation, with adaptations to the multi-agent setting.
- The approach relies on stronger external models for optimization and evaluation. Notably, the self-optimization setting performs even worse than the original model, raising concerns about whether the improvements mainly come from the stronger optimizer rather than the proposed method itself.
- There are concerns about the experimental setup. The reported baseline scores are significantly lower than official or commonly reported results. For example, the Qwen3-8B technical report reports 97.4 on MATH-500 (thinking mode) and 87.4 in non-thinking mode, while this paper reports only 74.8, and the proposed optimization method achieves 78.4. Similarly, DeepSeek-V3 reports 90.2 on MATH-500 in official reports, whereas this paper reports 81.5 after the proposed optimization. Similar discrepancies appear across other datasets and models. The paper should clarify the inference setup and explain these gaps.
- The paper does not report inference token usage or cost. Since reasoning benchmarks are highly sensitive to inference token budgets, it would be important to compare methods under a similar inference budget to ensure fair evaluation.
- Given the modest gains and the additional complexity of the optimization framework, it remains unclear whether the practical benefits justify the added cost and system complexity.

---

> ### Author Rebuttal · Authors · 2026-03-30
>
> Thank you for the review.
> > **Weakness 1:** Limited novelty
>
> We appreciate your assessment regarding the novelty. We respectfully argue that MASPO's core contribution lies in identifying and systematically addressing the Local-Global Misalignment problem unique to MAS, a structural challenge that cannot be resolved by directly combining single-agent optimization techniques. Each component of MASPO is specifically designed around this core problem.
>
> Other reviewers recognized the novelty of this work: Reviewer VApu considered it a "novel and important research question," and Reviewer YbbX noted the problem is "genuinely harder than single-agent prompt tuning" with a "reasonably well structured" method. These assessments support the research value of our problem formulation and framework design.
>
> > **Weakness 2:** Improvement attribution to stronger optimizer
>
> We appreciate your insightful question regarding the source of performance improvements. We believe there may be a misunderstanding. As shown in Table 2 (Part IV), the self-optimized setting (Qwen3-8B as both optimizer and evaluator) improves average accuracy from the Sequential MAS baseline of 65.31 to 67.70, demonstrating that MASPO's methodology itself is the fundamental driver of improvement, not merely the stronger external model. The additional gain from using Gemini-2.5-Pro (70.39) is expected, as a stronger optimizer naturally produces higher-quality candidate prompts and more reliable preference judgments.
>
> > **Weakness 3:** Baseline score discrepancies
>
> We sincerely appreciate your careful examination of the baseline scores.Taking Qwen3 as an example, three key differences in our experimental setup explain the discrepancy, all of which are deliberate design choices for the MAS prompt optimization setting:
>
> 1.The Qwen3 report uses temperature=0.7 (non-thinking mode), while we set temperature=0 for more deterministic comparison across prompt configurations.
>
> 2.The Qwen3 report samples 64 responses per problem and reports the accuracy (pass@k-style), which naturally inflates scores. We use single-pass inference, better reflecting real deployment and true prompt quality.
>
> 3.The Qwen3 report allows 32,768 output tokens, while our MAS setting limits each agent to 4,096 tokens to reserve context window for cascaded multi-agent outputs.
>
> Crucially, all baselines are evaluated under **identical inference configurations**, ensuring fair comparison. MASPO's improvements under this unified setup are reliable and reproducible.
>
> > **Weakness 4 & 5**: Inference cost and practical justification
>
> We appreciate your practical concerns regarding inference cost and the cost-benefit trade-off. MASPO optimizes prompt content only, the MAS topology and number of agent calls remain identical before and after optimization. We report the average per-sample inference token usage across all datasets:
> | Method         | Average Tokens | Accuracy (Avg.) |
> |----------------|-------------|-----------------|
> | Sequential MAS | 3,457       | 65.31           |
> | + TPE          | 5,098       | 66.49           |
> | + SPO          | 5,712       | 66.56           |
> | + MASPO        | 6,125       | 70.39           |
>
> The token increase stems from optimized prompts eliciting more thorough reasoning (e.g., independent verification, structured analysis), consistent with test-time compute scaling rather than redundant overhead.
>
> Regarding optimization cost, MASPO incurs a **one-time cost**, and the optimized prompts can be reused indefinitely. We report average optimized cost across all datasets:
> | Method | Average Cost | Optimization Time |
> |--------|-----------|-------------------|
> | SPO    | $2.4      | 1.9 hours         |
> | MASPO  | $15.6     | 2.7 hours         |
>
> As shown in Table 3, prompts optimized on Qwen3-8B transfer effectively to other models with consistent gains, meaning a single optimization investment yields returns across multiple models and deployment scenarios.

---

> > ### Author Rebuttal · Reviewer_p8hr · 2026-04-02
> >
> > I appreciate the authors’ response, which clarified many of my concerns.
> > For Weakness 2, thanks for clarification; it would be helpful to add a Sequential MAS row in Table 2, as the base method is currently unclear and I used Hierarchical MAS for comparison.
> > Regarding the mismatched scores, I believe the numbers I referred to are acc rather than pass@64, and this discrepancy could be discussed in the paper; the notably undertuned baseline configuration (eg, about −10 difference) remains a concern. The added results on inference and optimization cost are useful, and I have adjusted my score accordingly.

---

> > > ### Author Response · Authors · 2026-04-03
> > >
> > > Thanks for your careful review and constructive follow-up suggestions. We address each point below.
> > > > 1.For Weakness 2, thanks for clarification; it would be helpful to add a Sequential MAS row in Table 2, as the base method is currently unclear and I used Hierarchical MAS for comparison.
> > >
> > > Thank you for this suggestion. As noted in Lines 326–327 of the paper, the analyses in Table 2 are conducted on the Sequential MAS architecture. We fully agree that explicitly including the Sequential MAS baseline row in Table 2 would make the comparison more self-contained and immediately clear. We will incorporate this in the revised version.
> > >
> > > > 2.Regarding the mismatched scores, I believe the numbers I referred to are acc rather than pass@64, and this discrepancy could be discussed in the paper; the notably undertuned baseline configuration (eg, about −10 difference) remains a concern.
> > >
> > > Thank you for pressing on this important point. We agree that the MATH-500 scores in the Qwen3 report are single-pass accuracy. To further validate MASPO's effectiveness under inference settings closer to the official configuration, we conducted an additional experiment using the Qwen3 report's parameters (temperature=0.7, top-p=0.8, top-k=20, presence_penalty=1.5), with the only difference being max tokens:
> > > | Method | MATH-500 (avg. of 3 runs) |
> > > |---|---|
> > > | Vanilla | 83.27 ± 0.25 |
> > > | Sequential MAS | 83.80 ± 0.16 |
> > > | + MASPO | **85.67 ± 0.58** |
> > >
> > > The results confirm that MASPO delivers consistent improvements under this configuration as well. The remaining gap to the reported 87.4 likely stems from our per-agent token budget versus the 32,768 tokens allowed in the official evaluation, as well as potential differences in other undisclosed implementation details (e.g., system prompt, post-processing).

---

### Decision · Program_Chairs · 2026-04-30

**Decision:**

Accept (regular)

**Comment:**

This paper proposes MASPO, a framework for jointly optimizing prompts in multi-agent LLM systems, aiming to address the local–global misalignment problem that arises when optimizing agents independently. The method combines a multi-granularity evaluation signal with misalignment-aware sampling and an evolutionary beam search procedure to optimize prompts across interacting agents.
Reviewers agree that the paper tackles an important and timely problem, and that the proposed framework is well-structured and practically relevant. The empirical evaluation is relatively broad, covering multiple tasks, agent topologies, and transfer settings, and the results show consistent improvements over several prompt optimization baselines. The rebuttal further strengthens the paper by providing additional experiments, including controlled comparisons under equal computational budgets, cost analysis, and qualitative examples illustrating how prompt optimization improves multi-agent coordination.

Several concerns were raised during the review process. First, multiple reviewers questioned the novelty of the work, noting that the method combines existing components such as evolutionary search, beam search, and LLM-based evaluation. Second, there were concerns about the experimental setup, particularly discrepancies between reported baseline performance and prior work. The authors clarified that these differences arise from deliberate choices in inference configuration, and they also provided additional experiments under settings closer to prior reports, which partially alleviates this concern. Third, the reliance on LLM-based evaluation raised questions about reliability; the authors addressed this by providing agreement analysis with human annotations and demonstrating robustness across different evaluators.

From my assessment, the paper presents a well-executed engineering framework with some practical value。 Therefore, I recommend reject.